# Scaling Laws for Floating–Point Quantization Training

Xingwu Sun [* 1 2]   Shuaipeng Li [* † 1]

Ruobing Xie [1]   Weidong Han [1]   Kan Wu [1]   Zhen Yang [1]   Yixing Li [1 3]   An Wang [1 4]   Shuai Li [1]   Jinbao Xue [1]
Yu Cheng [3]   Yangyu Tao [1]   Zhanhui Kang [1]   Chengzhong Xu [† 2]   Di Wang [† 1]   Jie Jiang [1]

## Abstract

Low-precision training is considered an effective strategy for reducing both training and downstream inference costs. Previous scaling laws for precision mainly focus on integer quantization, which pay less attention to the constituents in floating-point (FP) quantization, and thus cannot well fit the LLM losses in this scenario. In contrast, while FP quantization training is more commonly implemented in production, it's research has been relatively superficial. In this paper, we thoroughly explore the effects of FP quantization targets, exponent bits, mantissa bits, and the calculation granularity of the scaling factor in FP quantization training performance of LLM models. In addition to an accurate FP quantization unified scaling law, we also provide valuable suggestions for the community: (1) Exponent bits contribute slightly more to the model performance than mantissa bits. We provide the optimal exponent-mantissa bit ratio for different bit numbers, which is available for future reference by hardware manufacturers; (2) We discover the formation of the critical data size in low-precision LLM training. Too much training data exceeding the critical data size will inversely bring in degradation of LLM performance; (3) The optimal FP quantization precision is directly proportional to the computational power, but within a wide computational power range. We estimate that the best cost-performance precision should lie between 4-8 bits.

[*]Equal contribution  [1]Tencent Hunyuan [2]University of Macau [3]The Chinese University of Hong Kong [4]Institute of Science Tokyo. Correspondence to: Shuaipeng Li <shuaipengli@tencent.com>, Chengzhong Xu <czxu@um.edu.mo>, Di Wang <diwang@tencent.com>.

*Proceedings of the 42nd International Conference on Machine Learning*, Vancouver, Canada. PMLR 267, 2025. Copyright 2025 by the author(s).

## 1. Introduction

Scaling laws of large language models (LLMs) could help developers effectively select superior parameter settings before experiments and accurately predict the model performance under different configurations. They are regarded as excellent guidance in LLM training. The widely acknowledged scaling law efforts such as Kaplan et al. (2020), Hoffmann et al. (2022), and Li et al. (2024) mainly concentrated on the central factors, i.e., model size and trained token size, which significantly impact the performance of LLMs. With the rapid growth of both model and data sizes, there has been increasing attention to the efficiency and cost of LLM training. Training and serving with lower precision becomes a popular solution. Currently, lots of representative LLMs were trained in BF16 and even lower precision (Dubey et al., 2024; Sun et al., 2024; Liu et al., 2024; Yang et al., 2024; Ma et al., 2024; Wang et al., 2023; Peng et al., 2023), aiming to balance effectiveness and efficiency. Compared to integer quantization, floating-point (FP) quantization can better maintain LLMs' accuracy at extremely lower bit rates and thus is often equipped in low-precision LLMs. Therefore, exploring the scaling laws of LLM performance under different low precision settings with FP quantization becomes essential to shed light on future low-precision LLM training.

Recently, there was pioneer work that conducted in-depth analyses and explorations on the LLM's scaling laws for precision in both training and inference (Kumar et al., 2024), quantitatively measuring the degradation rules of post-train quantization and quantized training. This scaling law provides an appropriate conclusion explaining the potential damage of excessively increasing training data to low-precision LLMs' performance. However, Kumar et al. (2024) directly adopted the bit width as the precision in its low-precision scaling laws, which might lose finer-grained modeling of the relationship between various parameter settings related to the FP quantization and the final loss of LLMs. In practice, the key factors of FP quantization such as the exponent, mantissa, and the block size of scaling factors may have different impacts on the final loss. A more comprehensive, precise, and practical scaling law for FP quantized training related to the data size (D), model size

(N), exponent (E), mantissa (M), and block size of scaling factors (B) is urgently desired.

Our work concentrates on establishing, verifying, and analyzing the scaling law for FP quantized training in LLMs. At the beginning, we first predict the model performance via the precision-related scaling law from previous work under different data/model sizes and precision settings. Surprisingly, we discover that the predictive performance was not perfectly satisfactory under different FP quantized training settings. Subsequently, we carefully design a comprehensive set of explorations with experiments of different precision settings (training 366 models), exploring the basic scaling law formation, as well as the potential impact of the quantization targets, exponent and mantissa, and block sizes on the loss. Finally, we aggregate these factors to get our final scaling law for FP quantized training with valuable insights to guide the LLM training under low precision.

Our FP quantization training scaling law, namely **Capybara** (Appendix B), is formulated as follows:

$$
\begin{aligned}
L(N, D, E, M, B) &= \frac{n}{N^\alpha} + \frac{d}{D^\beta} + \epsilon \\
&+ \frac{D^\beta}{N^\alpha} \frac{\log_2 B}{\gamma(E+0.5)^\delta(M+0.5)^\nu}.
\end{aligned}
\tag{1}
$$

The first two factors $D$ and $N$ indicate the data size and model size respectively, which show the main impacts on training loss given by the key factors of data and model size similar to the Chinchilla scaling law (Hoffmann et al., 2022); $\epsilon$ represents the bias; The last factor could be regarded as the additional negative impact deriving from low precision training, where $\frac{D^\beta}{N^\alpha}$ implies a certain form of "knowledge intensity" in LLM, and $\log_2 B$, $(E+0.5)^\delta$, and $(M+0.5)^\nu$ jointly reflect the "low precision information loss" of FP quantized training. We have conducted extensive fitting experiments with various possible scaling law formulations to ensure the accuracy and simplicity of our scaling laws. Note that the exponential hyper-parameters $\alpha$ and $\beta$ of model and data sizes are exactly the same as those in the first two factors. The product of the above "knowledge intensity" and "low precision information loss" forms the last factor.

Figure 1 illustrates the fitting results of our Capybara scaling law compared with other ones, demonstrating our advantages on predicting LLM performances under different float quantized training settings. Throughout our experiments and analyses related to our Capybara scaling law, we also discover the following observations and insights that could facilitate future low-precision LLM training: (a) It has been discovered that the impact of quantized weights on the performance is relatively minor during both forward and backward computations. Meanwhile, activations demonstrate a higher degree of quantization tolerance specifically when computing gradients pertaining to themselves. (b) The data

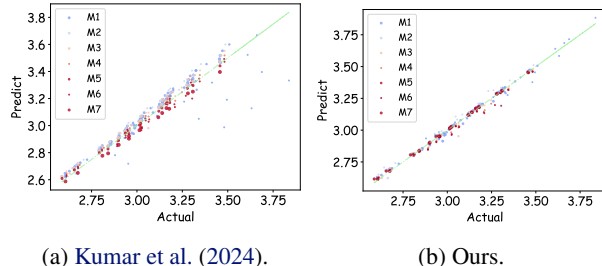

(a) Kumar et al. (2024).          (b) Ours.

*Figure 1.* Comparing Eq. (6) from Kumar et al. (2024) and Eq. (13) from our work, our Capybara scaling law fits data better in low - precision scenarios. Specifically, Kumar et al. (2024)'s fitting results show considerable bias in the E1M1 case. In the subfigures, data point magnitudes are roughly proportional to $E$.

size of LLM pre-training cannot be added indefinitely without harming the performance under low precision, while large model sizes, higher precision settings (measured by exponent and mantissa), and smaller block sizes could increase the extreme point of effective trained tokens for LLM training. (c) Intuitively, the negative impact of low-precision training in LLMs would be proportionally amplified with the "knowledge intensity". (d) The exponent and mantissa have their optimal settings under different bit widths. Exponent bits contribute slightly more to the model performance than mantissa bits. (e) The optimal FP quantization precision exhibits a direct proportionality with computational power. Nonetheless, across a broad spectrum of computational power, our estimated optimal cost-performance precision should reside within the 4-8 bits range.

## 2. Preliminary

**Classical Scaling Laws.** Scaling laws have become a fundamental framework for understanding the relationship between essential factors such as model size (N), data size (D), and the resulting loss (L) in deep learning. Two classical scaling laws have been widely recognized in the industry: Chinchilla scaling law (Hoffmann et al., 2022) and OpenAI scaling law (Kaplan et al., 2020). The Chinchilla scaling law is expressed as:

$$
L(N, D) = \frac{n}{N^\alpha} + \frac{d}{D^\beta} + \epsilon.
\tag{2}
$$

The OpenAI scaling law is given by:

$$
L(N, D) = \left[\left(\frac{n}{N}\right)^{\frac{\alpha}{\beta}} + \frac{d}{D}\right]^\beta + \epsilon,
\tag{3}
$$

where $n$, $d$, $\alpha$, $\beta$, and $\epsilon$ are positive fitted constants. The balance between $N$ and $D$ emerges as critical for compute-optimal training.

**Scaling Laws for Precision.** Subsequent research extends this framework by incorporating the role of precision in

quantized training and inference, so as to provide insights into how precision affects model performance. In Kumar et al. (2024), precision-aware scaling laws were introduced to capture the trade-offs between model size $N$, data size $D$, and precision $P$. For integer quantized training, they proposed the tradeoff between weight $N$ and weight precision $P$ as:

$$N_{\text{eff}}(N, P) = N(1 - e^{-P/\gamma}), \qquad (4)$$

where $N_{\text{eff}}$ indicates the "effective parameter count" of models, and $\gamma$ is a constant representing the sensitivity of model weights to precision. Incorporating $N_{\text{eff}}$ into the Chinchilla scaling law yields:

$$L(N, D, P) = \frac{n}{[N(1 - e^{-P/\gamma})]^\alpha} + \frac{d}{D^\beta} + \epsilon. \qquad (5)$$

This framework highlights that reducing weight precision $P$ can be compensated by increasing the parameter count $N$ to maintain performance, which is a critical insight for low-precision model optimization.

**Current Scaling Laws cannot Fit Well in FP Quantization.** Note that most previous work focused on integer quantized training. FP quantization is more prevalent in real-world applications due to its hardware compatibility and finer granularity. For instance, formats such as FP16 and BF16 are standard in many large-scale training pipelines, and emerging formats like FP8 and FP4 are gaining traction. Despite this, scaling laws specifically tailored to FP quantization are still largely unexplored. The primary distinction between FP and integer quantization lies in the allocation and usage of bits. FP numbers allocate bits to represent both the exponent and the mantissa, with each set of bits serving distinct purposes: the exponent mainly captures dynamic range, while the mantissa mainly encodes precision within that range. In contrast, integer formats uniformly distribute all bits to refine the quantization lattice, providing consistent resolution across the representable range. This fundamental difference highlights the need for dedicated scaling laws for the unique characteristics of FP formats.

Kumar et al. (2024) hypothesized that the exponent and mantissa bits should be scaled jointly (i.e., increase together as total bit count does). Then, in FP formats, precision is determined by the exponent $E$ and mantissa $M$, with the total precision: $P = E + M + 1$. By substituting $P$ in their precision-aware scaling law, we have:

$$L(N, D, E, M) = \frac{n}{\left[N(1 - e^{-\frac{1+E+M}{\gamma}})\right]^\alpha} + \frac{d}{D^\beta} + \epsilon, \quad (6)$$

However, upon conducting experiments and applying this scaling law to fit empirical results, particularly for low-precision training regimes, we observed significant deviations between the law's predictions and actual performance,

as illustrated in Figure 1a. The unsatisfactory fit, especially for training results using low-bit FP formats, suggests that the previous relationship proposed in Kumar et al. (2024) does not adequately capture the nuanced dynamic impacts of FP quantization on LLM performance.

In this work, we address these shortcomings by re-deriving the scaling law for FP quantized training. Our re-derivation incorporates a more nuanced understanding of how the finer factors of exponent, mantissa, and block size affect low-precision training. By refining the theoretical framework and aligning it more closely with observed behaviors, we aim to establish a more accurate and predictive scaling law so as to bridge the gap between theoretical insights and real-world applications.

## 3. Setup and Scaling Laws

### 3.1. Method and Implementation

**Quantization Method.** We quantized a tensor into a low-precision FP format, following the IEEE 754 standard (Kahan, 1996), which includes both normal and subnormal representations. The format consists of a sign bit, $E$ exponent bits and $M$ mantissa bits. To expand the dynamic range, the special bits are adopted for normal values instead of representing Infinity and Not a Number (NaN). Since the modern hardware does not support arbitrary FP format, we simulate them using QPyTorch (Zhang et al., 2019) with nearest rounding. Due to the narrow dynamic range and low representation precision of the low-precision format, we employ scaling techniques (Sun et al., 2019; Micikevicius et al., 2022). The original tensor is multiplied by a higher-precision scaler before being cast into the low-precision format. The scaling factor is computed as follows:

$$S_i = \text{FP}_{\max}/\max\left(|\mathbf{X}_{[\mathbf{B}i:\mathbf{B}(i+1)]}|\right), \qquad (7)$$

where $\text{FP}_{\max}$ represents the maximum normal value of the low-precision FP format. A scaling factor can be shared every $B$ elements along the channel dimension. It is a unified representation for tensor-wise scaling ($B = bd_{\text{in}}$), channel-wise scaling ($B = d_{\text{in}}$) and block-wise ($1 \leq B < d_{\text{in}}$) scaling for a tensor with the shape of $b \times d_{\text{in}}$.

**Implementation.** The quantization is applied to the linear layers in transformer (Vaswani et al., 2017), excluding the dot-product attention and the classifier. In a linear layer's computation, there is one matrix multiplication during the forward phase and two during the backward phase as:

$$\begin{cases} \mathbf{Y} & = \mathbf{X}\mathbf{W}^T \\ d\mathbf{X} & = d\mathbf{Y}_1\mathbf{W}_{\text{bwd}} \\ d\mathbf{W} & = (d\mathbf{Y}_2)^T\mathbf{X}_{\text{bwd}}, \end{cases} \qquad (8)$$

where $\mathbf{X} \in \mathbb{R}^{b \times d_{\text{in}}}$, $\mathbf{W} \in \mathbb{R}^{d_{\text{out}} \times d_{\text{in}}}$ and $\mathbf{Y} \in \mathbb{R}^{b \times d_{\text{out}}}$ rep-

resent the input tensor, the weight matrix and the output tensor, respectively. $b$ denotes the batch size, while $d_{\mathrm{in}}$ and $d_{\mathrm{out}}$ refer to the number of input and output channels. The two inputs per matrix multiplication are converted into a low-precision format with scaling factors. The inputs are de-quantized into BF16 tensors (Abadi et al., 2016), and BF16 multiplication is performed. The accumulators are stored in FP32 format, and the result of the accumulators are converted into a BF16 tensor as the output.

**Modeling Object.** As shown in Eq. (37) of (Kuzmin et al., 2022) , using the Signal-to-Quantization-Noise Ratio (SQNR) for GEMM operations could unify the effects of exponent (E) and mantissa (M) precision while accounting for input distribution modifications like random orthogonal transformations. However, SQNR calculation inherently depends on tensor distributions, which may limit its practical applicability. Therefore, we directly select the raw exponent and mantissa bits rather than the SNR as essential factors in our Capybara scaling law for more precise prediction ability.

## 3.2. Setup

We trained and evaluated a range of LLaMA (Dubey et al., 2024) architecture models on a subset of the Dolma V1.7 dataset (Soldaini et al., 2024), using the same sampling proportion as for the OLMo 7B-v1.7 model (Groeneveld et al., 2024). Our experiments systematically explored language model pretraining across $N \in \{41, 85, 154, 679\}$ million parameters and $D \in \{10, 20, 50, 100\}$ billion tokens. Furthermore, we conducted two additional pretraining sessions with 1.2 billion-parameter models to validate our Capybara scaling law equation. For every $(N, D)$ combination, we ran over 36 experiments, systematically varying exponents and mantissas, adjusting the quantization target during training, and exploring different block sizes for quantization. In total, we carried out 366 runs. Detailed hyperparameters and ablation studies are provided in Table 1 and Table 3.

## 3.3. Basic Scaling Law Form

Our research, building on foundational scaling laws for training data and model size that are crucial for understanding machine learning models' efficiency and effectiveness, first evaluates the classical Chinchilla (Hoffmann et al., 2022) and OpenAI (Kaplan et al., 2020) scaling laws and then presents our precision - aware design. To find which scaling law better fits empirical data, we ran experiments with BF16 precision for model sizes from 41 million to 679 million parameters and plotted scaling law curves to show the fit between predicted and actual losses. Since the Chinchilla scaling law had a better fit in our BF16 precision experiments (as shown in Figure 8), we used it as the basis for exploring our proposed float precision scaling law. Our aim

is to expand the understanding of scaling laws to include the impact of numerical precision on model performance, especially focusing on the trade - offs among precision, computational efficiency, and model accuracy.

In the following, we will present our methodology for investigating float - precision scaling laws, experimental results, and implications for model design and training at different numerical precision levels. Experiments will be used to discuss the marginal effects of exponent (E), mantissa (M), and scaling factor size (B) on LLM performance.

## 3.4. Quantization Targets

In our pursuit of balancing practicality with academic rigor, we have chosen to focus on the quantization of inputs to the General Matrix Multiplication (GEMM) computations within the Transformer architecture. Transformer consists of three main GEMM operations: forward computation, input gradient computation, and parameter gradient computation. The inputs to these matrix multiplications in both forward and backward passes include six distinct elements: $\mathbf{X}$, $\mathbf{W}$, $\mathrm{d}\mathbf{Y}_1$, $\mathbf{W}_{\mathrm{bwd}}$, $\mathrm{d}\mathbf{Y}_2$, and $\mathbf{X}_{\mathrm{bwd}}$, which can be quantized to $(P_1)$ through $(P_6)$ respectively (see Figure 2a).

**Experimental Findings.** Figure 2a illustrates the classical quantization targets of P1 to P6 to be explored. Through our experiments, we observed that the FP3 quantization of these inputs has varying impacts on LLM results. As illustrated in Figure 2b, our key observations related to quantization targets are as follows: (1) The quantization of P1, P3, and P5 has significant effects on the model's performance, leading to a substantial increase in loss. Notably, the quantization of P5 results in a pronounced degradation of performance, with losses increasing by up to 2%. This suggests that compressing and losing information in the input embedding during the backward pass could lead to considerable performance penalties. (2) Quantizing only one target of P4 or P6 yields the optimal performance. (3) Quantizing both P2 and P6 together results in similar overall performance to quantizing P2 alone. Quantizing P2, P4, and P6 together also leads to overall results comparable to quantizing P2 alone.

**Optimized Quantization Target.** To balance efficiency and effectiveness, we've selected P2, P4, and P6 for quantization. Future research will build on this setup, chosen for its minimal impact on performance compared to other options. This approach preserves model integrity while gaining computational benefits.

## 3.5. Exponent and Mantissa

Exponents and mantissas are key elements in FP representations. Proper bit-width assignments for them can significantly mitigate information loss during FP quantization. Here, we aim to explore the hidden associations between

(a) Quantization Targets.

(b) Loss Gaps.

*Figure 2.* For the subsequent exploration of scaling laws, we choose P2, P4, and P6 as our quantization targets.

exponents/mantissas and LLM performance.

**Exponent.** Firstly, we investigate the scaling law when Exponent serves as an independent variable. Experiments of different Exponents with various other parameter settings have been conducted, followed by our attempt of parameter fitting. It is assumed that the Exponent-related scaling law conforms more to a power-law relationship form as:

$$L(E) = \frac{\gamma}{(E + 0.5)^\delta} + \iota. \qquad (9)$$

We discussed other forms of relationships (e.g., Kumar et al. (2024)) and conducted relevant comparative experiments, ultimately finding that the power-law relationship is more consistent with the experimental results. The 0.5 in Eq. (9) functions as a good bias to fit extreme values. According to the IEEE 754 standard (Kahan, 1996), when either E (exponent) or M (mantissa) is set to 0, a default information is still retained. This can also account for the existence of a half-bit bias here.

Next, we attempt to fuse the relationship of E with those of data size D and model size N. Parameter fitting is carried out under various E, D, and N configurations, and the results are shown in Figure 10, where $\gamma$ is negatively correlated with $\frac{D}{N}$, and $\iota$ is negatively correlated with both $N$ and $D$. With regard to $\gamma$, we re-parameterize it as a function of $\frac{D^\phi}{\gamma N^\eta}$. As for $\iota$, simply multiplying the original form of the chinchilla scaling law by a coefficient precisely satisfies the pattern we discovered. Subsequently, we fit $L(N, D, E)$, which is the

jointly scaling law of N, D, E, as follows:

$$L(N, D, E) = \frac{D^\phi}{N^\eta} \frac{1}{\gamma(E + 0.5)^\delta} + \iota L_{BF16}. \qquad (10)$$

$L_{BF16}$ represents the BF16 loss $L(N, D)$ given in the chinchilla scaling law of Eq. (2), i.e, $L_{BF16} = \frac{n}{N^\alpha} + \frac{d}{D^\beta} + \epsilon$. Elegantly, we find that $\phi \approx \beta$, $\eta \approx \alpha$, and $\iota \approx 1$ in the fitting. Therefore, the Exponent scaling law is as:

$$L(N, D, E) = \frac{D^\beta}{N^\alpha} \frac{1}{\gamma(E + 0.5)^\delta} + \frac{n}{N^\alpha} + \frac{d}{D^\beta} + \epsilon. \quad (11)$$

Finally, we re-fit the data using Eq 11, obtaining the results shown in Figure 9a.

**Mantissa.** Similar to Exponent, we assume the Mantissa-related scaling law conforms a power-law form as $L(M) = \frac{\gamma'}{(M + 0.5)^\nu} + \iota'$. Surprisingly, jointly considering the effects of D and N in representing $\gamma'$ and $\iota'$, we also find that $\phi' \approx \beta$, $\eta' \approx \alpha$, and $\iota' \approx 1$. Ultimately, we adopt the form of Eq. (12) to fit the joint scaling law of Mantissa with $N$ and $D$.

$$L(N, D, M) = \frac{D^\beta}{N^\alpha} \frac{1}{\gamma(M + 0.5)^\nu} + \frac{n}{N^\alpha} + \frac{d}{D^\beta} + \epsilon. \quad (12)$$

The fitting result is shown in Figure 9b.

**Joint Exponent & Mantissa.** Integrating the scaling law results of Exponent and Mantissa when each serves as an independent variable, we can naturally organize their joint

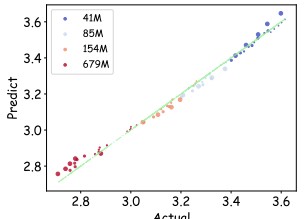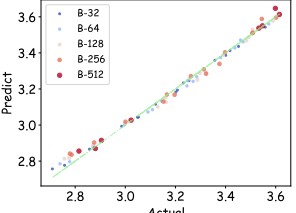

*Figure 3.* Our scaling law of Eq. (15) precisely forecasts validation loss for diverse block sizes. Data point sizes are directly proportional to $D$ and $B$ in the respective left and right sub-figures.

scaling law with N, D, E, M into the form of Eq. (13). The final fitting effect is presented in Figure 1b.

$$L(N, D, E, M) = \frac{D^\beta}{N^\alpha} \frac{1}{\gamma(E+0.5)^\delta(M+0.5)^\nu} + L_{BF16}.$$
(13)

### 3.6. Block Size of Scaling Factor

In this subsection, we discuss the correlations of the block sizes and LLM losses, and extend the block-related scaling law to channel-wise and tensor-wise strategies.

**Block-wise Strategy.** In the quantization process, we define the statistical range of the scaling factor as the block size (B). Since the scaling factor employs high-precision caching, when $B = 1$, it is equivalent to retaining a high-precision copy of the tensor to be quantized. At this point, the model's expressiveness should be approximately the same as that of the high-precision model: $L(N, D, B = 1) \approx L_{BF16}(N, D)$. After comparing the power, linear, and logarithmic law forms, we ultimately select the following logarithmic form as the scaling law when block size serves as an independent variable:

$$L(B) = \kappa \log_2 B + \psi.$$
(14)

In Figure 11, we demonstrate the changes in fitted $\kappa$ and $\psi$ under different N and D conditions. It can be observed that, similar to the exploration of Exponent and Mantissa in Section 3.5, $\kappa$ is positively correlated with $\frac{D}{N}$, while $\psi$ is negatively correlated with $N$ and $D$, respectively. Furthermore, after re-parameterizing them, we similarly found that the fitted exponents of $N$ and $D$ are approximately $\alpha$ and $\beta$, and the correction coefficient of $\psi$ based on the chinchilla scaling law is approximately equal to 1. Therefore, we ultimately build the scaling law for block size in conjunction with $N$ and $D$ as follows:

$$L(N, D, B) = \frac{D^\beta}{N^\alpha} \frac{\log_2 B}{\kappa} + \frac{n}{N^\alpha} + \frac{d}{D^\beta} + \epsilon.$$
(15)

The fitting effect on experimental data is shown in Figure 3.

**Channel-wise Strategy.** To investigate the scaling law under the channel-wise strategy, we first utilize Eq. (15) to inversely derive the equivalent block size for different N and D cases that achieve the same validation loss as when employing the channel-wise strategy. It is found that this equivalent block size of the channel-wise strategy is approximately a constant: $\log_2 B_{channel} \approx 13.1567$, which is natural since the batch size of gradient is (mostly) much larger than the hidden size ($d_{in}$ in Section 3.1). After incorporating this equivalent block size into Eq. (15), the fitted scenario of the channel-wise strategy is in Figure 13a.

**Tensor-wise Strategy.** Using the tensor-wise strategy, we apply Eq. (15) to predict the equivalent block size. As shown in Figure 12, this size follows a power-law relationship: $\log_2 B_{tensor} \approx \frac{N^\omega}{\xi D^\eta}$. Substituting this prediction into Eq. (15) produces the fitted results in Figure 13b.

## 4. A Unified Scaling Law for Floating–Point Quantization Training

In this section, we provide the unified scaling law for FP quantization training, with its fitting and predictive performance as well as the insightful findings deriving from our Capybara scaling law.

### 4.1. The Unified Scaling Law Formation

The unified scaling law for FP quantization training should be able to jointly consider all factors of the data size (D), model size (N), exponent (E), mantissa (M), and block size of scaling factors (B) for precise low-precision LLM performance prediction. Based on the conclusions drawn in Eq. (2), Eq. (13) and Eq. (15), we could intuitively design the unified scaling law as follows:

$$L(N, D, E, M, B) = \frac{n}{N^\alpha} + \frac{d}{D^\beta} + \epsilon + \rho(E, M, B).$$
(16)

Here, $\frac{n}{N^\alpha} + \frac{d}{D^\beta} + \epsilon$ represents the classical BF16 loss of the Chinchilla scaling law, and $\rho(E, M, B)$ indicates the additional negative impacts brought by low-precision float quantized training. Deriving from Eq. (13) and (15), we can formulate as follows:

$$\rho(E, M, B) = \frac{D^\beta}{N^\alpha} \frac{\log_2 B}{\gamma(E+0.5)^\delta(M+0.5)^\nu},$$
(17)

where the exponent E, mantissa M, and block size B jointly represent the possible information loss of FP quantized training, and $\frac{D^\beta}{N^\alpha}$ reflects, in a sense, the knowledge intensity of an LLM of size N trained on D data. Note that we could smoothly fuse the factors of E, M, and B with a unified set of hyper-parameters of $\alpha$, $\beta$, and $\gamma$.

We adopted all above 358 experiments in Section 3 containing various N, D, E, M, B settings to obtain the specific

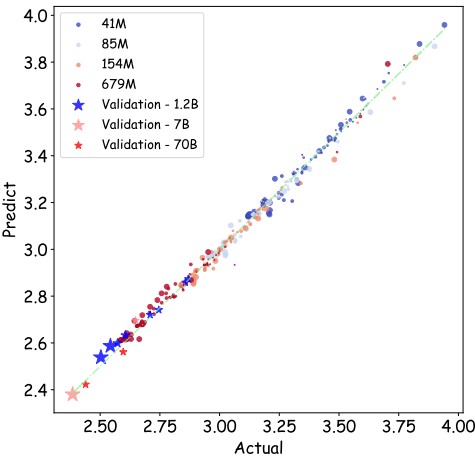

*Figure 4.* The fitting results of our Capybara scaling law for FP quantization training. Data point size is proportional to $D$. The star points (1.2B, 7B and 70B models) are our validation.

hyper-parameters in Eq. (16) and Eq. (17).

To ensure the simplicity and universality of our Capybara scaling law, we selected Occam's Razor to fuse or expurgate unnecessary hyper-parameters. Ultimately, the final scaling law for FP quantization training is articulated as follows:

$$L(N, D, E, M, B) = \frac{n}{N^\alpha} + \frac{d}{D^\beta} + \epsilon \\ + \frac{D^\beta}{N^\alpha} \frac{\log_2 B}{\gamma(E + 0.5)^\delta(M + 0.5)^\nu}, \quad (18)$$

where the corresponding hyper-parameters are given in Table 2. The fitting performances of Eq. (26) are given in Figure 4, which show superior capability compared to previous scaling laws in low-precision training.

Furthermore, we evaluate our Capybara scaling law to predict the losses of 1.2B, 7B and 70B LLMs with different low-precision settings and trained tokens (which are viewed as our validation models that are not used in calculating hyper-parameters of our Capybara scaling law). The consistently accurate fitting results demonstrate that our Capybara scaling law performs well in larger model sizes.

### 4.2. Implication-1: Optimal Float Layout Analysis

The optimal float layout for a given precision $P = E + M + 1$ is derived to minimize the impact of precision-related information loss on model performance. Based on Eq. (26), the optimal mantissa is expressed as:

$$M_{\text{opt}} = \frac{\nu P}{\delta + \nu} - 0.5. \quad (19)$$

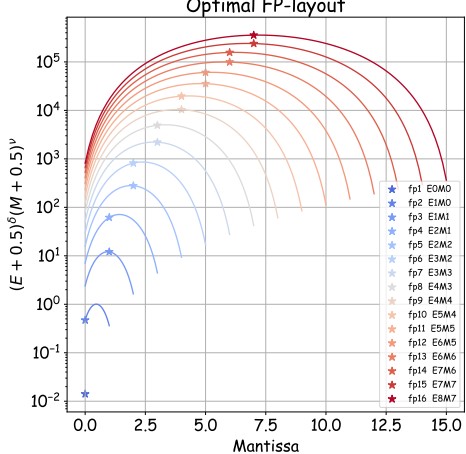

*Figure 5.* The optimal float layouts of different bit widths.

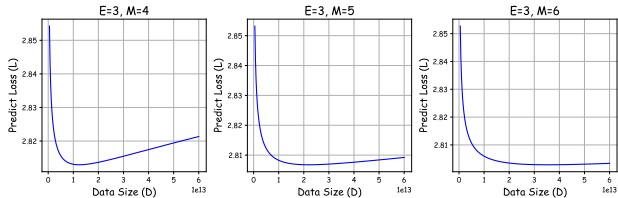

*Figure 6.* Variation of loss with data size under different FP quantization settings.

More details are in Appendix C. The corresponding loss scaling equation incorporates this optimal layout as:

$$L(N, D, P, B) = \frac{n}{N^\alpha} + \frac{d}{D^\beta} + \epsilon + \frac{D^\beta}{N^\alpha} \cdot \frac{\log_2 B}{\gamma_\rho P^{\delta+\nu}}, \quad (20)$$

where $\gamma_\rho = \frac{\gamma \delta^\delta \nu^\nu}{(\delta+\nu)^{\delta+\nu}}$. Figure 5 visualizes the predictive performance for different $P$, demonstrating the effectiveness of the derived layout in preserving performance under varying precisions. The optimal float layouts of FP4, FP8, and FP16 are E2M1, E4M3, and E8M7 (BF16), respectively.

### 4.3. Implication-2: Critical Data Size for Optimal Performance

From Eq. (26) and Figure 6, we can observe that there are two factors that contain $D$ and they have opposite impacts on LLM loss. Intuitively, it implies that there is an optimal data size under a certain FP quantization setting. The determination of critical data size ($D_{crit}$) stands as a critical juncture within the quantized training regimen. Upon exceeding the threshold of $D_{crit}$ with the training dataset, any additional data introduction negatively impacts the model efficacy, manifesting in a rise in validation loss instead of a decline. A comprehensive derivation for the estimation of

$D_{crit}$ is delineated in Appendix D:

$$D_{\text{crit}} = \left[ \frac{d\gamma N^\alpha (E+0.5)^\delta (M+0.5)^\nu}{\log_2 B} \right]^{\frac{1}{2\beta}}. \quad (21)$$

Notably, a positive correlation emerges between model size ($N$) or training precision ($P$) and the occurrence of this pivotal point, indicating its delayed emergence under such conditions. Utilizing our parameter estimation framework, a 1 billion-parameter model trained utilizing BF16 exhibits a $D_{crit}$ value of 1730T, which is much larger than our current data size, elucidating the previous lack of observation of this phenomenon. Conversely, when the same model is trained with FP8-E4M3, the $D_{crit}$ value swiftly diminishes to 27T, and with FP4-E2M1, it further plummets to 0.4T. This phenomenon implies the potential harmness of larger data size on low-precision LLM training.

### 4.4. Implication-3: Compute-Optimality with Fixed Configurations

We control the total computation cost $C = kNPD$ and analysis the optimal configuration under Eq. (26).

**Fixed Data Size $D$.** For a fixed $D$, the optimal precision $P_{\text{crit}}$ minimizes the loss while accounting for computational constraints. From Appendix E.1, $P_{\text{crit}}$ is expressed as:

$$P_{\text{opt}}(D) = \left( \gamma_D D^\beta \log_2 B \right)^{\frac{1}{\delta+\nu}}, \quad (22)$$

where $\gamma_D = \frac{\delta+\nu-\alpha}{n\alpha\gamma_\rho}$, which consolidates the relationships between model precision and compute efficiency. Eq. (22) suggests that we can adopt such a quantization strategy: in the early stage of training, employ aggressive quantization strategies such as FP8-E4M3, or even FP4-E2M1 which may be available in the future hardware, so as to quickly converge the model to a better level. Subsequently, as the data volume as well as the "knowledge intensity" further increase, gradually enhance the training precision to BF16, or even FP32, in order to maintain the optimal cost-effectiveness of training.

**Fixed Model Size $N$.** For a fixed $N$, the optimal precision depends on balancing the compute resources and maintaining the required training data size. The corresponding scaling law and derivations are in Appendix E.2. We have:

$$P_{\text{opt}}(N) = \left[ \gamma_N \left( \frac{C}{k} \right)^{2\beta} N^{-(\alpha+2\beta)} \log_2 B \right]^{\frac{1}{\delta+\nu+2\beta}}. \quad (23)$$

A finding analogous to that presented in Kumar et al. (2024) is observed herein, specifically, under the limitation of computational resources, an equilibrium exists between precision and model size from Eq. (23) is as:

$$P_{\text{opt}}^{\delta+\nu+2\beta} N^{\alpha+2\beta} = \text{Constant}. \quad (24)$$

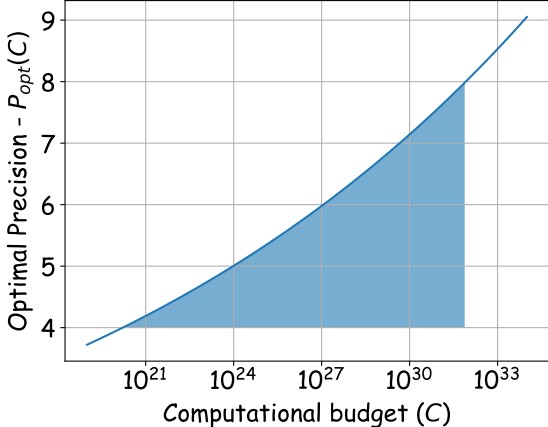

*Figure 7.* The optimal cost-performance ratio precision as a function of the total compute budget, illustrating the relationship between precision ($P$) and computational budget ($C$) when the block size ($B$) is set to 128 and $k = 6/16$.

**Minimization over $N$, $D$, $P$ with Fixed Compute.** Through a joint analysis of the impacts of $N$, $D$, and $P$ on the final validation loss, the relationship between cost-effective precision and expected compute budget is as:

$$P_{\text{opt}}^{(\delta+\nu)\frac{\alpha+\beta}{\beta}+\alpha} = \lambda \left( \gamma_D \log_2 B \right)^{\frac{\alpha+\beta}{\beta}} \left( \frac{C}{k} \right)^\alpha, \quad (25)$$

where $\lambda = \frac{d\beta}{n\alpha} \cdot \frac{\delta+\nu-\alpha}{\delta+\nu+\beta}$. In Appendix E.3, we present more detailed derivation processes. Based on the parameters fitted from our experimental data presented in Table 2, with the block size ($B$) set to 128 and $k = 6/16$, as illustrated in Figure 7, when the total compute budget is in the range of $(10^{21}, 10^{31})$ FP operations, the optimal cost-performance ratio precision is found to lie between 4 and 8 bits. This finding implies that training larger models with lower precision and utilizing less data yields a more cost-effective approach. Developers could rely on our implications from our Capybara scaling law to decide their optimal float-pointing quantization settings.

## 5. Related Work

**Scaling Law of LLMs.** Due to the extremely large cost of resource and time for LLM training, discovering appropriate scaling laws to accurately predict LLM capabilities under different parameters is essential for product-level training. Kaplan et al. (2020) gave the classic form of the scaling law and concludes that the performance penalty depends predictably on the ratio $N^{0.74}/D$. Hoffmann et al. (2022) modeled the final loss as a parametric function of the count of model parameters and the number of trained tokens, i.e. $E + A/N^\alpha + B/D^\beta$. Bahri et al. (2024) and Lin et al. (2024) theoretical analysis of how loss scales with the size of the

training dataset and the number of parameters in a power-law manner. Previous work (Dettmers & Zettlemoyer, 2023) explored the scaling laws of LLMs with different bit precisions. Recently, Kumar et al. (2024) focused on the impact of model parameters and data volume with low precision, highlighting the possible negative impacts of more trained tokens in low-precision LLM training and serving. Other recent work (Ouyang et al., 2024) also investigated the correlations of integer quantization and LLM performance. Nevertheless, it is still not that particularly clear to conclude the scaling law for FP quantization training with respect to the specific selection of the exponent, mantissa, and block size of scaling factors.

**Quantization of LLMs.** The quantization technique of large language models (Lang et al., 2024; Shen et al., 2024) has received widespread attention. Xiao et al. (2023) reduced the accuracy loss during quantization by smoothing the distribution of activations and weights. Dettmers et al. (2024) combined Quantization-Aware Training and LoRA methods to implement an efficient fine-tuning method. Egiazarian et al. (2024) explored techniques for compressing large language models at very low bit rates, and Behdin et al. (2023) proposed a framework that allows each layer to be quantized independently. Although previous work (Zhang et al., 2023; Yoshida, 2023) studied the impact of exponent, mantissa and blocksize on the quantification of LLMs, the comprehensive impact of these indicators has not been systematically studied and summarized.

## 6. Conclusion, Limitation, and Future Work

In this work, we propose our Capybara scaling law, which functions satisfactorily as a precise guidance of future low-precision LLM training. The key factors of the data size (D), model size (N), exponent (E), mantissa (M), and block size of scaling factors (B) are carefully considered in our Capybara scaling law throughout several hundred of experiments with various precision and model settings. Besides the scaling law, we also discover some insightful implications that could instruct and enhance future FP quantization training in LLMs. We hope our findings could shed light on better low-prediction LLM training to facilitate the LLM community.

In the future, we will verify our scaling laws for FP quantization training under larger model sizes and data sizes. Currently, our explorations are conducted based on the classical Transformer architecture. Whether our scaling laws could also be smoothly applied to LLMs of other architectures (e.g., Mamba series (Dao & Gu, 2024)) is worth confirming. Moreover, our experiments focus on the classical FP quantization strategies, while other new-proposed low-bit LLM quantization methods and hardware should also be covered in the future.

## Acknowledgement

Ruobing Xie is supported by the Young Elite Scientists Sponsorship Program by CAST (2023QNRC001).

## Impact Statement

This paper presents work whose goal is to advance the field of Machine Learning. There are many potential societal consequences of our work, none which we feel must be specifically highlighted here.

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

# A. Hyperparameter Details

The detailed hyper-parameters of our LLMs are given as follows:

*Table 1.* Model hyper-parameters for each size.

| Hyper-parameters | 41M | 85M | 154M | 679M | 1.2B | 7B | 70B |
|---|---|---|---|---|---|---|---|
| Layers | 12 | 12 | 12 | 24 | 24 | 35 | 88 |
| Hidden Size | 512 | 768 | 1024 | 1536 | 2048 | 4096 | 8192 |
| FFN Hidden Size | 1536 | 2048 | 2816 | 4096 | 5632 | 10752 | 22016 |
| Attention Heads | 8 | 12 | 16 | 24 | 32 | 32 | 64 |
| Attention Head size | 64 | 64 | 64 | 64 | 64 | 64 | 64 |
| Optimizer | | | | AdamW | | | |
| Adam $(\beta_1, \beta_2)$ | | | | (0.9, 0.95) | | | |
| Adam $\epsilon$ | | | | $1 \times 10^{-8}$ | | | |
| Weight Decay | | | | 0.1 | | | |
| Clip Grad Norm | | | | 1.0 | | | |
| Max LR | | | | $3.0 \times 10^{-4}$ | | | |
| Min LR | | | | 0 | | | |
| LR Decay | | | | Cosine | | | |
| Decay Rate | | | | 10% | | | |
| Seqence Length | | | | 2048 | | | |
| Batch Size (# Tokens) | | | | 2M | | | |
| Warmup Steps | | | | 500 | | | |

# B. Fitting Details of Our Scaling Laws

We name our scaling law for Floating-point quantization training as the Capybara scaling law. Under constrained resources and space, increasing the number of capybaras can significantly reduce their survival rate and quantity once a certain density threshold is surpassed. We observe a similar phenomenon in our scaling law: with a fixed model size, expanding the data size does not consistently yield improvements when the "knowledge density" becomes too high under the pressure of low-precision training. More fitting details of our Capybara scaling laws as introduced as follows.

## B.1. Numberical Fits

Our Capybara scaling law is formalized as:

$$L(N, D, E, M, B) = \frac{n}{N^\alpha} + \frac{d}{D^\beta} + \epsilon + \frac{D^\beta}{N^\alpha} \frac{\log_2 B}{\gamma(E + 0.5)^\delta (M + 0.5)^\nu}, \tag{26}$$

where the detailed fitted constants and values are shown as follows:

*Table 2.* Fitted constants and their values in Eq. (26).

| Constant | Value |
|---|---|
| $n$ | 69.2343 |
| $\alpha$ | 0.2368 |
| $d$ | 68973.0621 |
| $\beta$ | 0.5162 |
| $\epsilon$ | 1.9061 |
| $\gamma$ | 11334.5197 |
| $\delta$ | 3.1926 |
| $\nu$ | 2.9543 |

## B.2. Fitting Results of Classical Scaling Laws

We give the fitting results of two classical scaling laws, i.e., the Chinchilla scaling law and the OpenAI scaling law, in Figure 8. We select the Chinchilla scaling law for reference considering its better fitting performance.

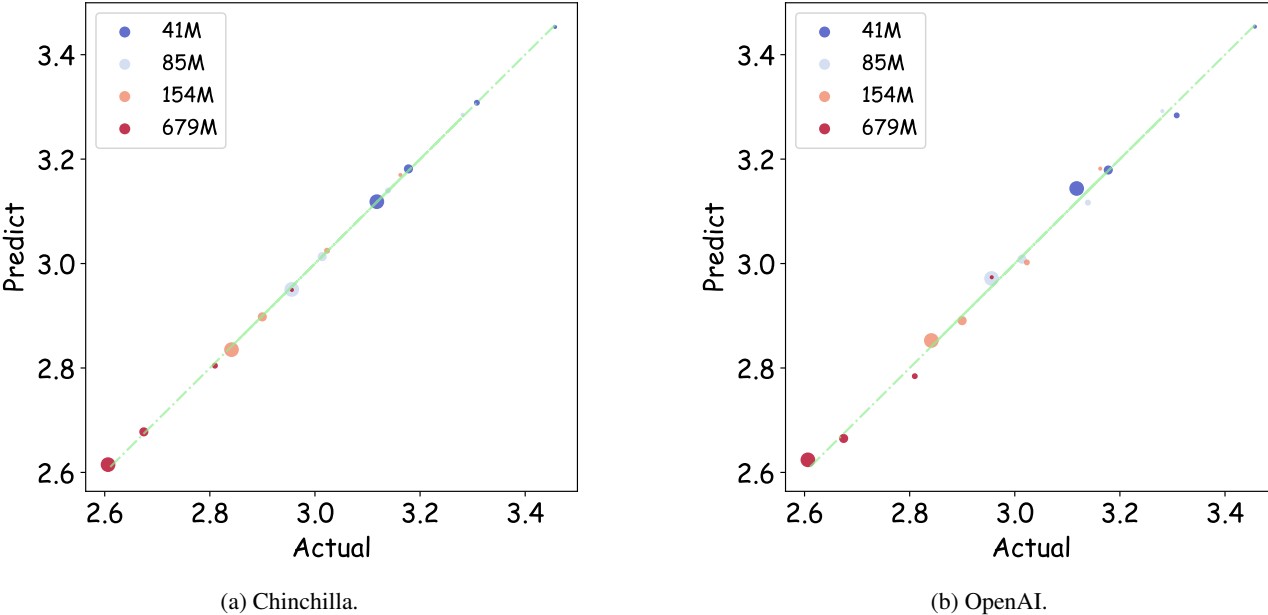

(a) Chinchilla.                              (b) OpenAI.

*Figure 8.* Fitting performance of classical scaling laws, with data point size proportional to $D$. Left: Curve based on Chinchilla scaling law shows excellent empirical training loss alignment with predicted losses. Right: Curve based on OpenAI scaling law also demonstrates a good match, though less precise than Chinchilla.

## B.3. Fitting Results of the Scaling Laws for Exponent and Mantissa

Figure 9 shows the fitting performance of our scaling law related to the exponent or mantissa, and Figure 10 shows the correlations between essential parameters in Eq. (9).

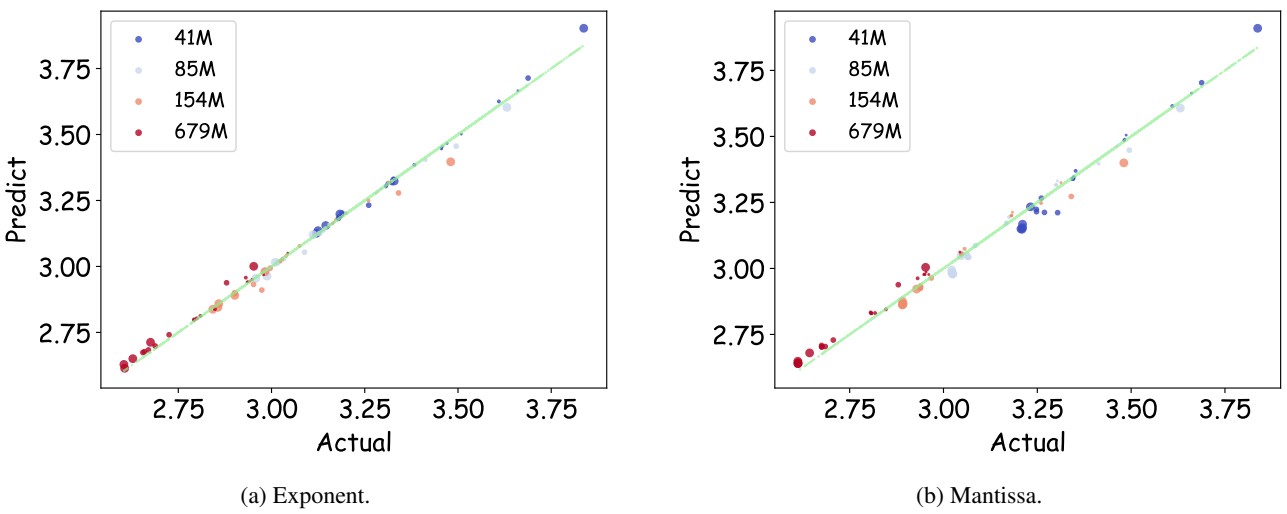

(a) Exponent.                              (b) Mantissa.

*Figure 9.* The fitting outcomes of our scaling law related to the exponent (left)/mantissa (right). Data point size is proportional to $D$.

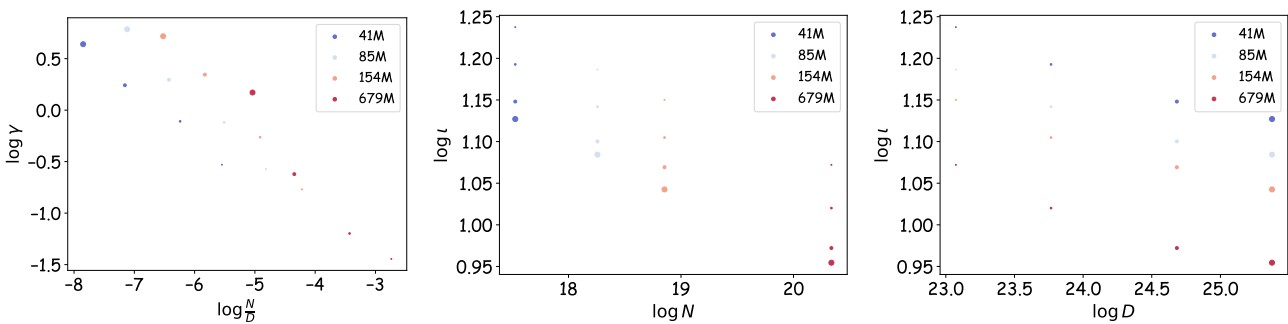

*Figure 10.* The correlations between $\gamma,\iota$ in Eq. (9) and $N,D$. $\gamma,\iota$ can be viewed as functions of $N,D$. Data point size is proportional to $D$.

## B.4. Fitting Results of the Scaling Laws for Scaling Factor

These figures show different correlations and fitting results related to the scaling factor B in our scaling law.

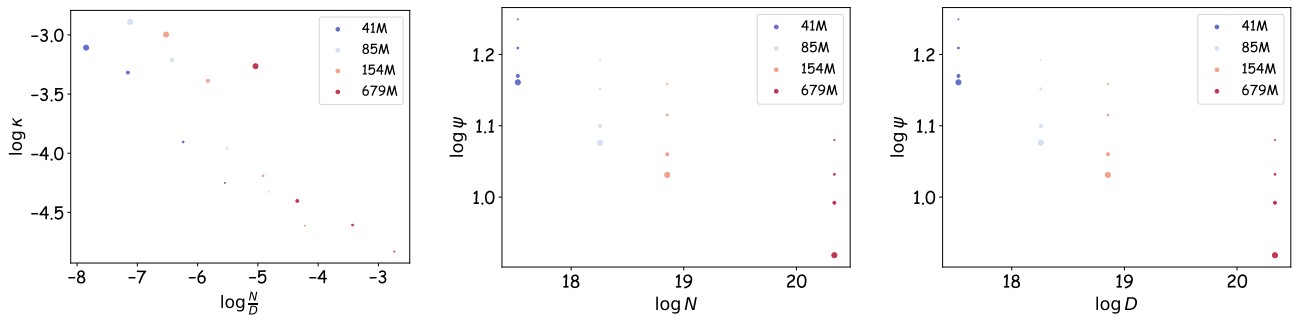

*Figure 11.* The correlations between $\kappa,\psi$ in Eq. (14) and $N,D$. $\kappa,\psi$ could be viewed as functions of $N,D$. The data points are scaled proportionally to the value of $D$.

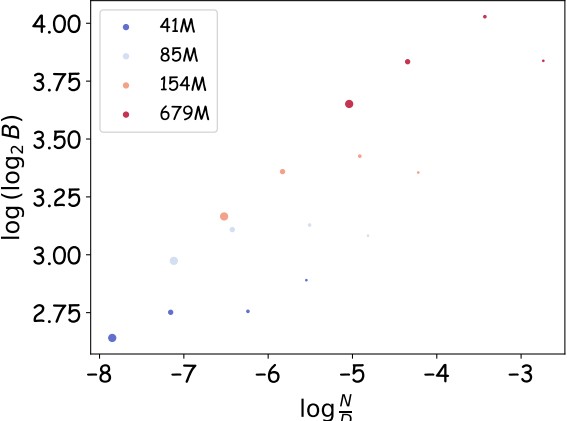

*Figure 12.* The correlations between $\log_2 B$ and $\frac{N}{D}$. The size of the data point is proportional to $D$.

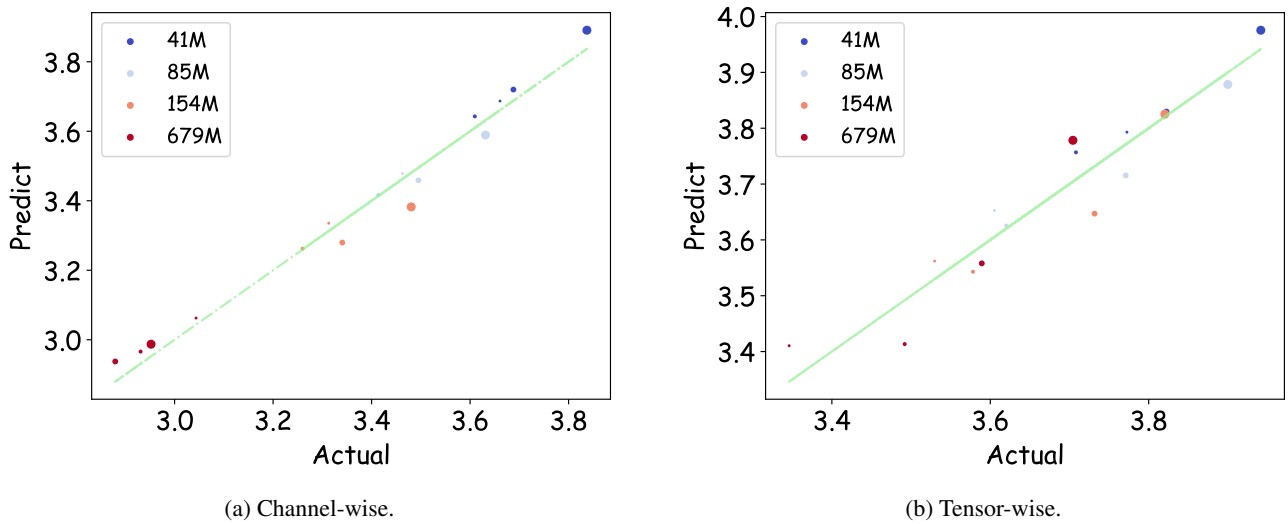

(a) Channel-wise.

(b) Tensor-wise.

*Figure 13.* Fitting results of the scaling law under different sharing strategies of the scaling factor. The size of the data point is proportional to $D$.

## C. Optimal Float Layout

Given a specified total precision (P), the process of determining the optimal allocation of exponent bits (E) and mantissa bits (M) involves substituting the equation:

$$E = P - M - 1. \tag{27}$$

into the proposed scaling law delineated in Eq. (17):

$$
\begin{aligned}
\rho(M) &= \frac{D^\beta}{N^\alpha} \cdot \frac{\log_2 B}{\gamma(P - 1 - M + 0.5)^\delta (M + 0.5)^\nu} \\
&= \frac{D^\beta}{N^\alpha} \cdot \frac{\log_2 B}{\gamma(P - 0.5 - M)^\delta (M + 0.5)^\nu}.
\end{aligned} \tag{28}
$$

Subsequently, the loss function $L$, with respect to the mantissa bits $M$, is expressed as:

$$L(M) = \frac{n}{N^\alpha} + \frac{d}{D^\beta} + \epsilon + \frac{D^\beta}{N^\alpha} \cdot \frac{\log_2 B}{\gamma(P - 0.5 - M)^\delta (M + 0.5)^\nu}. \tag{29}$$

To optimize this function, we compute the partial derivative of $L$ with respect to $M$:

$$
\begin{aligned}
\frac{\partial L}{\partial M} &= -\frac{D^\beta}{N^\alpha} \cdot \frac{\log_2 B}{\gamma} \frac{\nu(P - 0.5 - M)^\delta (M + 0.5)^{\nu-1} - \delta(P - 0.5 - M)^{\delta-1}(M + 0.5)^\nu}{(P - 0.5 - M)^{2\delta}(M + 0.5)^{2\nu}} \\
&= -\frac{D^\beta}{N^\alpha} \cdot \frac{\log_2 B}{\gamma} \cdot \frac{1}{(P - 0.5 - M)^\delta (M - 0.5)^\nu} \left( \frac{\nu}{M + 0.5} - \frac{\delta}{P - 0.5 - M} \right).
\end{aligned} \tag{30}
$$

By setting this partial derivative equal to zero, we obtain the optimal value for $M$ which is given by Eq. (31):

$$\frac{\partial L}{\partial M} = 0$$

$$\frac{\nu}{M + 0.5} - \frac{\delta}{P - 0.5 - M} = 0 \tag{31}$$

$$M = \frac{\nu P}{\delta + \nu} - 0.5,$$

and the corresponding value for $E$ is:

$$
\begin{aligned}
E_{opt} &= P - 1 - M_{opt} \\
&= \frac{\delta P}{\delta + \nu} - 0.5.
\end{aligned}
\tag{32}
$$

Next, we introduce the optimal values $M_{opt}$ and $E_{opt}$ into the proposed scaling law, as formulated in Eq. (17):

$$
\begin{aligned}
\rho_{opt}(P) &= \frac{D^\beta}{N^\alpha} \cdot \frac{\log_2 B}{\gamma \left(\frac{\delta P}{\delta + \nu}\right)^\delta \left(\frac{\nu P}{\delta + \nu}\right)^\nu} \\
&= \frac{D^\beta}{N^\alpha} \cdot \frac{\log_2 B}{\gamma} \cdot \frac{(\delta + \nu)^{\delta + \nu}}{\delta^\delta \nu^\nu} \cdot \frac{1}{P^{\delta + \nu}}.
\end{aligned}
\tag{33}
$$

For the sake of simplification, we introduce the parameter $\gamma_\rho$, defined as follows:

$$\gamma_\rho = \frac{\gamma \delta^\delta \nu^\nu}{(\delta + \nu)^{\delta + \nu}}. \tag{34}$$

As a result,

$$\rho_{opt}(P) = \frac{D^\beta}{N^\alpha} \cdot \frac{\log_2 B}{\gamma_\rho P^{\delta + \nu}}. \tag{35}$$

By substituting Eq. (35) into the unified scaling equation, namely Eq. (16), we arrive at Eq. (20), which is further simplified to:

$$
\begin{aligned}
L_{opt}(P) &= \frac{n}{N^\alpha} + \frac{d}{D^\beta} + \epsilon + \frac{D^\beta}{N^\alpha} \cdot \frac{\log_2 B}{\gamma_\rho P^{\delta + \nu}} \\
&= \frac{n}{N^\alpha} \left(1 + \frac{D^\beta}{n} \cdot \frac{\log_2 B}{\gamma_\rho P^{\delta + \nu}}\right) + \frac{d}{D^\beta} + \epsilon \\
&= \frac{n}{N^\alpha \left[\left(\frac{1}{1 + \frac{D^\beta \log_2 B}{n \gamma_\rho P^{\delta + \nu}}}\right)^{\frac{1}{\alpha}}\right]^\alpha} + \frac{d}{D^\beta} + \epsilon.
\end{aligned}
\tag{36}
$$

Furthermore, let $\gamma_n$ denote a constant value:

$$\gamma_n = \gamma_\rho n. \tag{37}$$

In accordance with the Chinchilla scaling law (Hoffmann et al., 2022), we define $N_{eff}$ as the count of effective parameters, which aligns with the model size specified in Eq. (2):

$$N_{eff} = N \left( \frac{1}{1 + \frac{D^\beta \log_2 B}{\gamma_n P^{\delta+\nu}}} \right)^{\frac{1}{\alpha}}. \tag{38}$$

When the condition $D^\beta \log_2 B \gg \gamma_n P^{\delta+\nu}$ is satisfied, the effective number of parameters, $N_{eff}$, can be simplified as follows:

$$N_{eff} \approx \left( \frac{\gamma_n}{D^\beta \log_2 B} \right)^{\frac{1}{\alpha}} \cdot NP^{\frac{\delta+\nu}{\alpha}}. \tag{39}$$

Hence, we discern a power-law relationship between the number of effective parameters, $N_{eff}$, and the precision, $P$. It is important to emphasize that $N_{eff}$ is influenced not solely by the quantization technique employed but also by the volume of data. When both the model size and the quantization method are held constant, an increase in data size leads to a decrease in the number of effective parameters.

## D. Critical Data Size

For FP quantization training, overtraining may occur, that is, when the amount of training data exceeds the critical value, the loss will increase instead. Given exponent bits (E), mantissa bits (M), block size (B), and number of model parameters (N), we aim to find the critical data size before over-training. Based on Eq. (16), we derive the expression for $D$ when the partial derivative with respect to $D$ is zero.

Preliminarily, we compute the partial derivative of the loss function $L$ with respect to the data size $D$:

$$\begin{aligned}
\frac{\partial L}{\partial D} &= \frac{\partial}{\partial D} \frac{n}{N^\alpha} + \frac{\partial}{\partial D} \frac{d}{D^\beta} + \frac{\partial}{\partial D} \epsilon + \frac{\partial}{\partial D} \rho(E, M, B) \\
&= -\beta \frac{d}{D^{\beta+1}} + \beta \frac{D^{\beta-1}}{N^\alpha} \cdot \frac{\log_2 B}{\gamma(E+0.5)^\delta (M+0.5)^\nu}.
\end{aligned} \tag{40}$$

By setting this partial derivative to zero and then solving for $D^\beta$, we obtain:

$$\frac{\partial L}{\partial D} = 0$$

$$-\beta \frac{d}{D^{\beta+1}} + \beta \frac{D^{\beta-1}}{N^\alpha} \cdot \frac{\log_2 B}{\gamma(E+0.5)^\delta (M+0.5)^\nu} = 0 \tag{41}$$

$$D^\beta = \sqrt{\frac{d\gamma N^\alpha (E+0.5)^\delta (M+0.5)^\nu}{\log_2 B}}.$$

Consequently, the critical value of $D$ is given by Eq. (21).

## E. Compute-optimality

In order to investigate the optimal precision $P$ under constrained computational budget, we define the computational expenditure associated with FP quantization training as follows:

$$C = k(P+b)ND. \tag{42}$$

Here, $k$ signifies a proportionality constant, $P$ denotes the computational cost per model parameter, and $b$ accounts for the additional expense incurred during the multiplication of scaling factors.

### E.1. Fixed data size $D$

For the critical precision $P$ in relation to the data size $D$, Eq. (42) can be incorporated into the proposed scaling law, as expressed in Eq. (20):

$$
\begin{aligned}
L(D, P) &= \frac{n}{\left(\frac{C}{k(P+b)D}\right)^\alpha} + \frac{d}{D^\beta} + \epsilon + \frac{D^\beta}{\left(\frac{C}{k(P+b)D}\right)^\alpha} \cdot \frac{\log_2 B}{\gamma_\rho P^{\delta+\nu}} \\
&= n\left(\frac{kD}{C}\right)^\alpha (P+b)^\alpha + \frac{D^\beta \log_2 B}{\gamma_\rho} \cdot \left(\frac{kD}{C}\right)^\alpha \cdot \frac{(P+b)^\alpha}{P^{\delta+\nu}} + \frac{d}{D^\beta} + \epsilon.
\end{aligned}
\tag{43}
$$

We then compute the partial derivative of the loss function $L(D, P)$ with respect to $P$:

$$
\begin{aligned}
\frac{\partial L(D, P)}{\partial P} &= n\alpha\left(\frac{kD}{C}\right)^\alpha (P+b)^{\alpha-1} + \frac{D^\beta \log_2 B}{\gamma_\rho} \cdot \left(\frac{kD}{C}\right)^\alpha \frac{\alpha(P+b)^{\alpha-1}P^{\delta+\nu} - (\delta+\nu)(P+b)^\alpha P^{\delta+\nu-1}}{P^{2(\delta+\nu)}} \\
&= n\alpha\left(\frac{kD}{C}\right)^\alpha (P+b)^{\alpha-1} + \frac{D^\beta \log_2 B}{\gamma_\rho} \cdot \left(\frac{kD}{C}\right)^\alpha \frac{(P+b)^\alpha}{P^{\delta+\nu}}\left(\frac{\alpha}{P+b} - \frac{\delta+\nu}{P}\right).
\end{aligned}
\tag{44}
$$

Upon setting the partial derivative of the loss function with respect to precision $P$ equal to zero, and solving for $P$, we obtain:

$$
\frac{\partial L(D, P)}{\partial P} = 0.
$$
$$
\frac{P+b}{P^{\delta+\nu}}\left(\frac{\delta+\nu}{P} - \frac{\alpha}{P+b}\right) = \frac{n\gamma_\rho\alpha}{D^\beta \log_2 B}.
\tag{45}
$$

Assuming

$$
\gamma_D = \frac{\delta+\nu-\alpha}{n\alpha\gamma_\rho},
\tag{46}
$$

and considering that $b = 0$, the critical precision $P$ is determined as:

$$
P^{\delta+\nu} = \gamma_D D^\beta \log_2 B.
\tag{47}
$$

As illustrated in Figure 14, Eq. (47) suggests that as the amount of training data increases, the most economical precision also correspondingly rises under the constraint of limited computational power.

### E.2. Fixed model size $N$

With respect to the critical precision $P$ in relation to the model size $N$, we can streamline Eq. (20) to:

$$
\begin{aligned}
L(N, P) &= \frac{n}{N^\alpha} + \frac{d}{\left(\frac{C}{k(P+b)N}\right)^\beta} + \epsilon + \frac{\left(\frac{C}{k(P+b)N}\right)^\beta}{N^\alpha} \cdot \frac{\log_2 B}{\gamma_\rho P^{\delta+\nu}} \\
&= d\left(\frac{kN}{C}\right)^\beta (P+b)^\beta + \frac{\log_2 B}{\gamma_\rho N^\alpha} \cdot \left(\frac{C}{kN}\right)^\beta \cdot \frac{1}{(P+b)^\beta P^{\delta+\nu}} + \frac{n}{N^\alpha} + \epsilon.
\end{aligned}
\tag{48}
$$

Subsequently, we evaluate the partial derivative of the loss function $L(N, P)$ with respect to $P$:

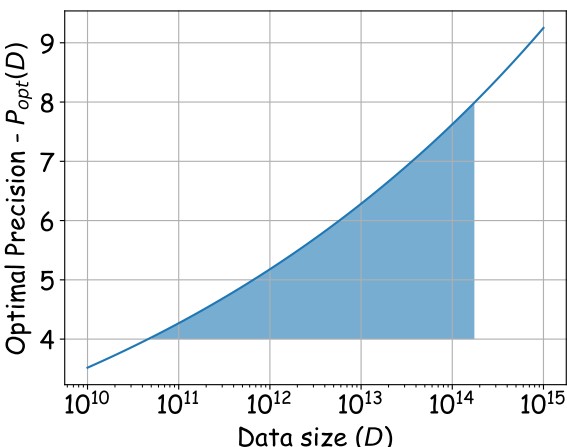

*Figure 14.* Under the constraint of computing the budget with block size ($B$) set to 128, and based on the results of our experimental data fitting, the optimal precision ($P$) values for different data sizes ($D$) can be deduced. As depicted, across a substantially broad range of data sizes from 0.1T to 100T, the optimal precision value consistently falls within the range of 4 to 8 bits.

$$
\begin{aligned}
\frac{\partial L(N,P)}{\partial P} &= d\beta \left(\frac{kN}{C}\right)^{\beta} (P+b)^{\beta-1} - \frac{\log_2 B}{\gamma_\rho N^\alpha} \cdot \left(\frac{C}{kN}\right)^{\beta} \cdot \frac{\beta(P+b)^{\beta-1}P^{\delta+\nu} + (\delta+\nu)(P+b)^{\beta}P^{\delta+\nu-1}}{(P+b)^{2\beta}P^{2(\delta+\nu)}} \\
&= d\beta \left(\frac{kN}{C}\right)^{\beta} (P+b)^{\beta-1} - \frac{\log_2 B}{\gamma_\rho N^\alpha} \cdot \left(\frac{C}{kN}\right)^{\beta} \cdot \frac{1}{(P+b)^{\beta}P^{\delta+\nu}} \left(\frac{\beta}{P+b} + \frac{\delta+\nu}{P}\right).
\end{aligned}
\tag{49}
$$

Upon setting this partial derivative to zero, and solving for $P$, we arrive at:

$$
\frac{\partial L(N,P)}{\partial P} = 0.
$$

$$
\frac{\log_2 B}{\gamma_\rho N^\alpha} \cdot \left(\frac{C}{kN}\right)^{\beta} \cdot \frac{1}{(P+b)^{\beta}P^{\delta+\nu}} \left(\frac{\beta}{P+b} + \frac{\delta+\nu}{P}\right) = d\beta \left(\frac{kN}{C}\right)^{\beta} (P+b)^{\beta-1}.
\tag{50}
$$

$$
\frac{1}{(P+b)^{2\beta-1}P^{\delta+\nu}} \left(\frac{\beta}{P+b} + \frac{\delta+\nu}{P}\right) = \frac{d\beta\gamma_\rho N^\alpha}{\log_2 B} \cdot \left(\frac{kN}{C}\right)^{2\beta}.
$$

By introducing

$$
\gamma_N = \frac{\beta+\delta+\nu}{d\beta\gamma_\rho},
\tag{51}
$$

and under the assumption that $b = 0$, the critical $P$ is deduced to be:

$$
\frac{1}{P^{2\beta-1}P^{\delta+\nu}} \left(\frac{\beta}{P} + \frac{\delta+\nu}{P}\right) = \frac{d\beta\gamma_\rho N^\alpha}{\log_2 B} \cdot \left(\frac{kN}{C}\right)^{2\beta}.
$$

$$
P^{\delta+\nu+2\beta} = \frac{(\beta+\delta+\nu)\log_2 B}{d\beta\gamma_\rho N^\alpha} \cdot \left(\frac{C}{kN}\right)^{2\beta}.
\tag{52}
$$

$$
P^{\delta+\nu+2\beta} = \gamma_N \left(\frac{C}{k}\right)^{2\beta} N^{-(\alpha+2\beta)} \log_2 B.
$$

### E.3. Minimization over $N$, $D$, $P$ with Fixed Compute

Based on the results from Section E.1, E.2 and specifically Eq. (42) with b=0, we proceed to address the system of equations:

$$\begin{cases} \frac{\partial L(D,P)}{\partial D} = 0. \\\\ \frac{\partial L(D,P)}{\partial P} = 0. \\\\ C = kPND. \end{cases} \tag{53}$$

This subsequently leads to the expression of $P$ in relation to the computational budget $C$:

$$P^{(\delta+\nu)\frac{\alpha+\beta}{\beta}+\alpha} = (\gamma_D \log_2 B)^{\frac{\alpha+\beta}{\beta}} \cdot \frac{d\beta}{n\alpha} \cdot \frac{\delta+\nu-\alpha}{\delta+\nu+\beta} \cdot \left(\frac{C}{k}\right)^{\alpha}. \tag{54}$$

## F. Detailed Settings of Experiments

We show the detailed configurations of our experiments as follows:

Table 3: All configurations for the ablation experiments.

|     | N        | D           | E   | M   | B       | Fitting support |
| --- | -------- | ----------- | --- | --- | ------- | --------------- |
| 0   | 40894464 | 10485760000 | 0   | 7   | channel | ✓               |
| 1   | 40894464 | 10485760000 | 1   | 1   | 32      | ✓               |
| 2   | 40894464 | 10485760000 | 1   | 1   | 64      | ✓               |
| 3   | 40894464 | 10485760000 | 1   | 1   | 128     | ✓               |
| 4   | 40894464 | 10485760000 | 1   | 1   | 256     | ✓               |
| 5   | 40894464 | 10485760000 | 1   | 1   | 512     | ✓               |
| 6   | 40894464 | 10485760000 | 1   | 1   | channel | ✓               |
| 7   | 40894464 | 10485760000 | 1   | 1   | tensor  | ✓               |
| 8   | 40894464 | 10485760000 | 1   | 2   | channel | ✓               |
| 9   | 40894464 | 10485760000 | 1   | 3   | channel | ✓               |
| 10  | 40894464 | 10485760000 | 1   | 4   | channel | ✓               |
| 11  | 40894464 | 10485760000 | 1   | 5   | channel | ✓               |
| 12  | 40894464 | 10485760000 | 1   | 6   | channel | ✓               |
| 13  | 40894464 | 10485760000 | 2   | 1   | channel | ✓               |
| 14  | 40894464 | 10485760000 | 2   | 3   | channel | ✓               |
| 15  | 40894464 | 10485760000 | 3   | 1   | channel | ✓               |
| 16  | 40894464 | 10485760000 | 3   | 2   | channel | ✓               |
| 17  | 40894464 | 10485760000 | 4   | 1   | channel | ✓               |
| 18  | 40894464 | 10485760000 | 4   | 3   | channel | ✓               |
| 19  | 40894464 | 10485760000 | 4   | 5   | channel | ✓               |
| 20  | 40894464 | 10485760000 | 5   | 1   | channel | ✓               |
| 21  | 40894464 | 10485760000 | 5   | 2   | channel | ✓               |
| 22  | 40894464 | 10485760000 | 6   | 1   | channel | ✓               |
| 23  | 40894464 | 20971520000 | 0   | 7   | channel | ✓               |
| 24  | 40894464 | 20971520000 | 1   | 1   | 32      | ✓               |
| 25  | 40894464 | 20971520000 | 1   | 1   | 64      | ✓               |
| 26  | 40894464 | 20971520000 | 1   | 1   | 128     | ✓               |
| 27  | 40894464 | 20971520000 | 1   | 1   | 256     | ✓               |
| 28  | 40894464 | 20971520000 | 1   | 1   | 512     | ✓               |
| 29  | 40894464 | 20971520000 | 1   | 1   | channel | ✓               |
| 30  | 40894464 | 20971520000 | 1   | 1   | tensor  | ✓               |
| 31  | 40894464 | 20971520000 | 1   | 2   | channel | ✓               |
| 32  | 40894464 | 20971520000 | 1   | 3   | channel | ✓               |

| | | | | | | |
|---|---|---|---|---|---|---|
| 33 | 40894464 | 20971520000 | 1 | 4 | channel | ✓ |
| 34 | 40894464 | 20971520000 | 1 | 5 | channel | ✓ |
| 35 | 40894464 | 20971520000 | 1 | 6 | channel | ✓ |
| 36 | 40894464 | 20971520000 | 2 | 1 | channel | ✓ |
| 37 | 40894464 | 20971520000 | 2 | 3 | channel | ✓ |
| 38 | 40894464 | 20971520000 | 3 | 1 | channel | ✓ |
| 39 | 40894464 | 20971520000 | 3 | 2 | channel | ✓ |
| 40 | 40894464 | 20971520000 | 4 | 1 | channel | ✓ |
| 41 | 40894464 | 20971520000 | 4 | 3 | channel | ✓ |
| 42 | 40894464 | 20971520000 | 4 | 5 | channel | ✓ |
| 43 | 40894464 | 20971520000 | 5 | 1 | channel | ✓ |
| 44 | 40894464 | 20971520000 | 5 | 2 | channel | ✓ |
| 45 | 40894464 | 20971520000 | 6 | 1 | channel | ✓ |
| 46 | 40894464 | 52428800000 | 0 | 7 | channel | ✓ |
| 47 | 40894464 | 52428800000 | 1 | 1 | 32 | ✓ |
| 48 | 40894464 | 52428800000 | 1 | 1 | 64 | ✓ |
| 49 | 40894464 | 52428800000 | 1 | 1 | 128 | ✓ |
| 50 | 40894464 | 52428800000 | 1 | 1 | 256 | ✓ |
| 51 | 40894464 | 52428800000 | 1 | 1 | 512 | ✓ |
| 52 | 40894464 | 52428800000 | 1 | 1 | channel | ✓ |
| 53 | 40894464 | 52428800000 | 1 | 1 | tensor | ✓ |
| 54 | 40894464 | 52428800000 | 1 | 2 | channel | ✓ |
| 55 | 40894464 | 52428800000 | 1 | 3 | channel | ✓ |
| 56 | 40894464 | 52428800000 | 1 | 4 | channel | ✓ |
| 57 | 40894464 | 52428800000 | 1 | 5 | channel | ✓ |
| 58 | 40894464 | 52428800000 | 1 | 6 | channel | ✓ |
| 59 | 40894464 | 52428800000 | 2 | 1 | channel | ✓ |
| 60 | 40894464 | 52428800000 | 2 | 3 | channel | ✓ |
| 61 | 40894464 | 52428800000 | 3 | 1 | channel | ✓ |
| 62 | 40894464 | 52428800000 | 3 | 2 | channel | ✓ |
| 63 | 40894464 | 52428800000 | 4 | 1 | channel | ✓ |
| 64 | 40894464 | 52428800000 | 4 | 3 | channel | ✓ |
| 65 | 40894464 | 52428800000 | 4 | 5 | channel | ✓ |
| 66 | 40894464 | 52428800000 | 5 | 1 | channel | ✓ |
| 67 | 40894464 | 52428800000 | 5 | 2 | channel | ✓ |
| 68 | 40894464 | 52428800000 | 6 | 1 | channel | ✓ |
| 69 | 40894464 | 104857600000 | 0 | 7 | channel | ✓ |
| 70 | 40894464 | 104857600000 | 1 | 1 | 32 | ✓ |
| 71 | 40894464 | 104857600000 | 1 | 1 | 64 | ✓ |
| 72 | 40894464 | 104857600000 | 1 | 1 | 128 | ✓ |
| 73 | 40894464 | 104857600000 | 1 | 1 | 256 | ✓ |
| 74 | 40894464 | 104857600000 | 1 | 1 | 512 | ✓ |
| 75 | 40894464 | 104857600000 | 1 | 1 | channel | ✓ |
| 76 | 40894464 | 104857600000 | 1 | 1 | tensor | ✓ |
| 77 | 40894464 | 104857600000 | 1 | 2 | channel | ✓ |
| 78 | 40894464 | 104857600000 | 1 | 3 | channel | ✓ |
| 79 | 40894464 | 104857600000 | 1 | 4 | channel | ✓ |
| 80 | 40894464 | 104857600000 | 1 | 5 | channel | ✓ |
| 81 | 40894464 | 104857600000 | 1 | 6 | channel | ✓ |
| 82 | 40894464 | 104857600000 | 2 | 1 | channel | ✓ |
| 83 | 40894464 | 104857600000 | 2 | 3 | channel | ✓ |
| 84 | 40894464 | 104857600000 | 3 | 1 | channel | ✓ |
| 85 | 40894464 | 104857600000 | 3 | 2 | channel | ✓ |
| 86 | 40894464 | 104857600000 | 4 | 1 | channel | ✓ |

| | | | | | | |
|---|---|---|---|---|---|---|
| 87 | 40894464 | 104857600000 | 4 | 3 | channel | ✓ |
| 88 | 40894464 | 104857600000 | 4 | 5 | channel | ✓ |
| 89 | 40894464 | 104857600000 | 5 | 1 | channel | ✓ |
| 90 | 40894464 | 104857600000 | 5 | 2 | channel | ✓ |
| 91 | 40894464 | 104857600000 | 6 | 1 | channel | ✓ |
| 92 | 84934656 | 10485760000 | 0 | 7 | channel | ✓ |
| 93 | 84934656 | 10485760000 | 1 | 1 | 32 | ✓ |
| 94 | 84934656 | 10485760000 | 1 | 1 | 64 | ✓ |
| 95 | 84934656 | 10485760000 | 1 | 1 | 128 | ✓ |
| 96 | 84934656 | 10485760000 | 1 | 1 | 256 | ✓ |
| 97 | 84934656 | 10485760000 | 1 | 1 | channel | ✓ |
| 98 | 84934656 | 10485760000 | 1 | 1 | tensor | ✓ |
| 99 | 84934656 | 10485760000 | 1 | 2 | channel | ✓ |
| 100 | 84934656 | 10485760000 | 1 | 3 | channel | ✓ |
| 101 | 84934656 | 10485760000 | 1 | 4 | channel | ✓ |
| 102 | 84934656 | 10485760000 | 1 | 5 | channel | ✓ |
| 103 | 84934656 | 10485760000 | 1 | 6 | channel | ✓ |
| 104 | 84934656 | 10485760000 | 2 | 1 | channel | ✓ |
| 105 | 84934656 | 10485760000 | 2 | 3 | channel | ✓ |
| 106 | 84934656 | 10485760000 | 3 | 1 | channel | ✓ |
| 107 | 84934656 | 10485760000 | 3 | 2 | channel | ✓ |
| 108 | 84934656 | 10485760000 | 4 | 1 | channel | ✓ |
| 109 | 84934656 | 10485760000 | 4 | 3 | channel | ✓ |
| 110 | 84934656 | 10485760000 | 4 | 5 | channel | ✓ |
| 111 | 84934656 | 10485760000 | 5 | 1 | channel | ✓ |
| 112 | 84934656 | 10485760000 | 5 | 2 | channel | ✓ |
| 113 | 84934656 | 10485760000 | 6 | 1 | channel | ✓ |
| 114 | 84934656 | 20971520000 | 0 | 7 | channel | ✓ |
| 115 | 84934656 | 20971520000 | 1 | 1 | 32 | ✓ |
| 116 | 84934656 | 20971520000 | 1 | 1 | 64 | ✓ |
| 117 | 84934656 | 20971520000 | 1 | 1 | 128 | ✓ |
| 118 | 84934656 | 20971520000 | 1 | 1 | 256 | ✓ |
| 119 | 84934656 | 20971520000 | 1 | 1 | channel | ✓ |
| 120 | 84934656 | 20971520000 | 1 | 1 | tensor | ✓ |
| 121 | 84934656 | 20971520000 | 1 | 2 | channel | ✓ |
| 122 | 84934656 | 20971520000 | 1 | 3 | channel | ✓ |
| 123 | 84934656 | 20971520000 | 1 | 4 | channel | ✓ |
| 124 | 84934656 | 20971520000 | 1 | 5 | channel | ✓ |
| 125 | 84934656 | 20971520000 | 1 | 6 | channel | ✓ |
| 126 | 84934656 | 20971520000 | 2 | 1 | channel | ✓ |
| 127 | 84934656 | 20971520000 | 2 | 3 | channel | ✓ |
| 128 | 84934656 | 20971520000 | 3 | 1 | channel | ✓ |
| 129 | 84934656 | 20971520000 | 3 | 2 | channel | ✓ |
| 130 | 84934656 | 20971520000 | 4 | 1 | channel | ✓ |
| 131 | 84934656 | 20971520000 | 4 | 3 | channel | ✓ |
| 132 | 84934656 | 20971520000 | 4 | 5 | channel | ✓ |
| 133 | 84934656 | 20971520000 | 5 | 1 | channel | ✓ |
| 134 | 84934656 | 20971520000 | 5 | 2 | channel | ✓ |
| 135 | 84934656 | 20971520000 | 6 | 1 | channel | ✓ |
| 136 | 84934656 | 52428800000 | 0 | 7 | channel | ✓ |
| 137 | 84934656 | 52428800000 | 1 | 1 | 32 | ✓ |
| 138 | 84934656 | 52428800000 | 1 | 1 | 64 | ✓ |
| 139 | 84934656 | 52428800000 | 1 | 1 | 128 | ✓ |
| 140 | 84934656 | 52428800000 | 1 | 1 | 256 | ✓ |

| | | | | | | |
|---|---|---|---|---|---|---|
| 141 | 84934656 | 52428800000 | 1 | 1 | channel | ✓ |
| 142 | 84934656 | 52428800000 | 1 | 1 | tensor | ✓ |
| 143 | 84934656 | 52428800000 | 1 | 2 | channel | ✓ |
| 144 | 84934656 | 52428800000 | 1 | 3 | channel | ✓ |
| 145 | 84934656 | 52428800000 | 1 | 4 | channel | ✓ |
| 146 | 84934656 | 52428800000 | 1 | 5 | channel | ✓ |
| 147 | 84934656 | 52428800000 | 1 | 6 | channel | ✓ |
| 148 | 84934656 | 52428800000 | 2 | 1 | channel | ✓ |
| 149 | 84934656 | 52428800000 | 2 | 3 | channel | ✓ |
| 150 | 84934656 | 52428800000 | 3 | 1 | channel | ✓ |
| 151 | 84934656 | 52428800000 | 3 | 2 | channel | ✓ |
| 152 | 84934656 | 52428800000 | 4 | 1 | channel | ✓ |
| 153 | 84934656 | 52428800000 | 4 | 3 | channel | ✓ |
| 154 | 84934656 | 52428800000 | 4 | 5 | channel | ✓ |
| 155 | 84934656 | 52428800000 | 5 | 1 | channel | ✓ |
| 156 | 84934656 | 52428800000 | 5 | 2 | channel | ✓ |
| 157 | 84934656 | 52428800000 | 6 | 1 | channel | ✓ |
| 158 | 84934656 | 104857600000 | 0 | 7 | channel | ✓ |
| 159 | 84934656 | 104857600000 | 1 | 1 | 32 | ✓ |
| 160 | 84934656 | 104857600000 | 1 | 1 | 64 | ✓ |
| 161 | 84934656 | 104857600000 | 1 | 1 | 128 | ✓ |
| 162 | 84934656 | 104857600000 | 1 | 1 | 256 | ✓ |
| 163 | 84934656 | 104857600000 | 1 | 1 | channel | ✓ |
| 164 | 84934656 | 104857600000 | 1 | 1 | tensor | ✓ |
| 165 | 84934656 | 104857600000 | 1 | 2 | channel | ✓ |
| 166 | 84934656 | 104857600000 | 1 | 3 | channel | ✓ |
| 167 | 84934656 | 104857600000 | 1 | 4 | channel | ✓ |
| 168 | 84934656 | 104857600000 | 1 | 5 | channel | ✓ |
| 169 | 84934656 | 104857600000 | 1 | 6 | channel | ✓ |
| 170 | 84934656 | 104857600000 | 2 | 1 | channel | ✓ |
| 171 | 84934656 | 104857600000 | 2 | 3 | channel | ✓ |
| 172 | 84934656 | 104857600000 | 3 | 1 | channel | ✓ |
| 173 | 84934656 | 104857600000 | 3 | 2 | channel | ✓ |
| 174 | 84934656 | 104857600000 | 4 | 1 | channel | ✓ |
| 175 | 84934656 | 104857600000 | 4 | 3 | channel | ✓ |
| 176 | 84934656 | 104857600000 | 4 | 5 | channel | ✓ |
| 177 | 84934656 | 104857600000 | 5 | 1 | channel | ✓ |
| 178 | 84934656 | 104857600000 | 5 | 2 | channel | ✓ |
| 179 | 84934656 | 104857600000 | 6 | 1 | channel | ✓ |
| 180 | 154140672 | 10485760000 | 0 | 7 | channel | ✓ |
| 181 | 154140672 | 10485760000 | 1 | 1 | 32 | ✓ |
| 182 | 154140672 | 10485760000 | 1 | 1 | 64 | ✓ |
| 183 | 154140672 | 10485760000 | 1 | 1 | 128 | ✓ |
| 184 | 154140672 | 10485760000 | 1 | 1 | 256 | ✓ |
| 185 | 154140672 | 10485760000 | 1 | 1 | channel | ✓ |
| 186 | 154140672 | 10485760000 | 1 | 1 | tensor | ✓ |
| 187 | 154140672 | 10485760000 | 1 | 2 | channel | ✓ |
| 188 | 154140672 | 10485760000 | 1 | 3 | channel | ✓ |
| 189 | 154140672 | 10485760000 | 1 | 4 | channel | ✓ |
| 190 | 154140672 | 10485760000 | 1 | 5 | channel | ✓ |
| 191 | 154140672 | 10485760000 | 1 | 6 | channel | ✓ |
| 192 | 154140672 | 10485760000 | 2 | 1 | channel | ✓ |
| 193 | 154140672 | 10485760000 | 2 | 3 | channel | ✓ |
| 194 | 154140672 | 10485760000 | 3 | 1 | channel | ✓ |

| 195 | 154140672 | 10485760000 | 3 | 2 | channel | ✓ |
| 196 | 154140672 | 10485760000 | 4 | 1 | channel | ✓ |
| 197 | 154140672 | 10485760000 | 4 | 3 | channel | ✓ |
| 198 | 154140672 | 10485760000 | 4 | 5 | channel | ✓ |
| 199 | 154140672 | 10485760000 | 5 | 1 | channel | ✓ |
| 200 | 154140672 | 10485760000 | 5 | 2 | channel | ✓ |
| 201 | 154140672 | 10485760000 | 6 | 1 | channel | ✓ |
| 202 | 154140672 | 20971520000 | 0 | 7 | channel | ✓ |
| 203 | 154140672 | 20971520000 | 1 | 1 | 32 | ✓ |
| 204 | 154140672 | 20971520000 | 1 | 1 | 64 | ✓ |
| 205 | 154140672 | 20971520000 | 1 | 1 | 128 | ✓ |
| 206 | 154140672 | 20971520000 | 1 | 1 | 256 | ✓ |
| 207 | 154140672 | 20971520000 | 1 | 1 | channel | ✓ |
| 208 | 154140672 | 20971520000 | 1 | 1 | tensor | ✓ |
| 209 | 154140672 | 20971520000 | 1 | 2 | channel | ✓ |
| 210 | 154140672 | 20971520000 | 1 | 3 | channel | ✓ |
| 211 | 154140672 | 20971520000 | 1 | 4 | channel | ✓ |
| 212 | 154140672 | 20971520000 | 1 | 5 | channel | ✓ |
| 213 | 154140672 | 20971520000 | 1 | 6 | channel | ✓ |
| 214 | 154140672 | 20971520000 | 2 | 1 | channel | ✓ |
| 215 | 154140672 | 20971520000 | 2 | 3 | channel | ✓ |
| 216 | 154140672 | 20971520000 | 3 | 1 | channel | ✓ |
| 217 | 154140672 | 20971520000 | 3 | 2 | channel | ✓ |
| 218 | 154140672 | 20971520000 | 4 | 1 | channel | ✓ |
| 219 | 154140672 | 20971520000 | 4 | 3 | channel | ✓ |
| 220 | 154140672 | 20971520000 | 4 | 5 | channel | ✓ |
| 221 | 154140672 | 20971520000 | 5 | 1 | channel | ✓ |
| 222 | 154140672 | 20971520000 | 5 | 2 | channel | ✓ |
| 223 | 154140672 | 20971520000 | 6 | 1 | channel | ✓ |
| 224 | 154140672 | 52428800000 | 0 | 7 | channel | ✓ |
| 225 | 154140672 | 52428800000 | 1 | 1 | 32 | ✓ |
| 226 | 154140672 | 52428800000 | 1 | 1 | 64 | ✓ |
| 227 | 154140672 | 52428800000 | 1 | 1 | 128 | ✓ |
| 228 | 154140672 | 52428800000 | 1 | 1 | 256 | ✓ |
| 229 | 154140672 | 52428800000 | 1 | 1 | channel | ✓ |
| 230 | 154140672 | 52428800000 | 1 | 1 | tensor | ✓ |
| 231 | 154140672 | 52428800000 | 1 | 2 | channel | ✓ |
| 232 | 154140672 | 52428800000 | 1 | 3 | channel | ✓ |
| 233 | 154140672 | 52428800000 | 1 | 4 | channel | ✓ |
| 234 | 154140672 | 52428800000 | 1 | 5 | channel | ✓ |
| 235 | 154140672 | 52428800000 | 1 | 6 | channel | ✓ |
| 236 | 154140672 | 52428800000 | 2 | 1 | channel | ✓ |
| 237 | 154140672 | 52428800000 | 2 | 3 | channel | ✓ |
| 238 | 154140672 | 52428800000 | 3 | 1 | channel | ✓ |
| 239 | 154140672 | 52428800000 | 3 | 2 | channel | ✓ |
| 240 | 154140672 | 52428800000 | 4 | 1 | channel | ✓ |
| 241 | 154140672 | 52428800000 | 4 | 3 | channel | ✓ |
| 242 | 154140672 | 52428800000 | 4 | 5 | channel | ✓ |
| 243 | 154140672 | 52428800000 | 5 | 1 | channel | ✓ |
| 244 | 154140672 | 52428800000 | 5 | 2 | channel | ✓ |
| 245 | 154140672 | 52428800000 | 6 | 1 | channel | ✓ |
| 246 | 154140672 | 104857600000 | 0 | 7 | channel | ✓ |
| 247 | 154140672 | 104857600000 | 1 | 1 | 32 | ✓ |
| 248 | 154140672 | 104857600000 | 1 | 1 | 64 | ✓ |

| | | | | | | |
|---|---|---|---|---|---|---|
| 249 | 154140672 | 104857600000 | 1 | 1 | 128 | ✓ |
| 250 | 154140672 | 104857600000 | 1 | 1 | 256 | ✓ |
| 251 | 154140672 | 104857600000 | 1 | 1 | channel | ✓ |
| 252 | 154140672 | 104857600000 | 1 | 1 | tensor | ✓ |
| 253 | 154140672 | 104857600000 | 1 | 2 | channel | ✓ |
| 254 | 154140672 | 104857600000 | 1 | 3 | channel | ✓ |
| 255 | 154140672 | 104857600000 | 1 | 4 | channel | ✓ |
| 256 | 154140672 | 104857600000 | 1 | 5 | channel | ✓ |
| 257 | 154140672 | 104857600000 | 1 | 6 | channel | ✓ |
| 258 | 154140672 | 104857600000 | 2 | 1 | channel | ✓ |
| 259 | 154140672 | 104857600000 | 2 | 3 | channel | ✓ |
| 260 | 154140672 | 104857600000 | 3 | 1 | channel | ✓ |
| 261 | 154140672 | 104857600000 | 3 | 2 | channel | ✓ |
| 262 | 154140672 | 104857600000 | 4 | 1 | channel | ✓ |
| 263 | 154140672 | 104857600000 | 4 | 3 | channel | ✓ |
| 264 | 154140672 | 104857600000 | 4 | 5 | channel | ✓ |
| 265 | 154140672 | 104857600000 | 5 | 1 | channel | ✓ |
| 266 | 154140672 | 104857600000 | 5 | 2 | channel | ✓ |
| 267 | 154140672 | 104857600000 | 6 | 1 | channel | ✓ |
| 268 | 679477248 | 10485760000 | 0 | 7 | channel | ✓ |
| 269 | 679477248 | 10485760000 | 1 | 1 | 32 | ✓ |
| 270 | 679477248 | 10485760000 | 1 | 1 | 64 | ✓ |
| 271 | 679477248 | 10485760000 | 1 | 1 | 128 | ✓ |
| 272 | 679477248 | 10485760000 | 1 | 1 | 256 | ✓ |
| 273 | 679477248 | 10485760000 | 1 | 1 | 512 | ✓ |
| 274 | 679477248 | 10485760000 | 1 | 1 | channel | ✓ |
| 275 | 679477248 | 10485760000 | 1 | 1 | tensor | ✓ |
| 276 | 679477248 | 10485760000 | 1 | 2 | channel | ✓ |
| 277 | 679477248 | 10485760000 | 1 | 3 | channel | ✓ |
| 278 | 679477248 | 10485760000 | 1 | 4 | channel | ✓ |
| 279 | 679477248 | 10485760000 | 1 | 5 | channel | ✓ |
| 280 | 679477248 | 10485760000 | 1 | 6 | channel | ✓ |
| 281 | 679477248 | 10485760000 | 2 | 1 | channel | ✓ |
| 282 | 679477248 | 10485760000 | 2 | 3 | channel | ✓ |
| 283 | 679477248 | 10485760000 | 3 | 1 | channel | ✓ |
| 284 | 679477248 | 10485760000 | 3 | 2 | channel | ✓ |
| 285 | 679477248 | 10485760000 | 4 | 1 | channel | ✓ |
| 286 | 679477248 | 10485760000 | 4 | 3 | channel | ✓ |
| 287 | 679477248 | 10485760000 | 4 | 5 | channel | ✓ |
| 288 | 679477248 | 10485760000 | 5 | 1 | channel | ✓ |
| 289 | 679477248 | 10485760000 | 5 | 2 | channel | ✓ |
| 290 | 679477248 | 10485760000 | 6 | 1 | channel | ✓ |
| 291 | 679477248 | 20971520000 | 0 | 7 | channel | ✓ |
| 292 | 679477248 | 20971520000 | 1 | 1 | 32 | ✓ |
| 293 | 679477248 | 20971520000 | 1 | 1 | 64 | ✓ |
| 294 | 679477248 | 20971520000 | 1 | 1 | 128 | ✓ |
| 295 | 679477248 | 20971520000 | 1 | 1 | 256 | ✓ |
| 296 | 679477248 | 20971520000 | 1 | 1 | 512 | ✓ |
| 297 | 679477248 | 20971520000 | 1 | 1 | channel | ✓ |
| 298 | 679477248 | 20971520000 | 1 | 1 | tensor | ✓ |
| 299 | 679477248 | 20971520000 | 1 | 2 | channel | ✓ |
| 300 | 679477248 | 20971520000 | 1 | 3 | channel | ✓ |
| 301 | 679477248 | 20971520000 | 1 | 4 | channel | ✓ |
| 302 | 679477248 | 20971520000 | 1 | 5 | channel | ✓ |

| | | | | | | |
|---|---|---|---|---|---|---|
| 303 | 679477248 | 20971520000 | 1 | 6 | channel | ✓ |
| 304 | 679477248 | 20971520000 | 2 | 1 | channel | ✓ |
| 305 | 679477248 | 20971520000 | 2 | 3 | channel | ✓ |
| 306 | 679477248 | 20971520000 | 3 | 1 | channel | ✓ |
| 307 | 679477248 | 20971520000 | 3 | 2 | channel | ✓ |
| 308 | 679477248 | 20971520000 | 4 | 1 | channel | ✓ |
| 309 | 679477248 | 20971520000 | 4 | 3 | channel | ✓ |
| 310 | 679477248 | 20971520000 | 4 | 5 | channel | ✓ |
| 311 | 679477248 | 20971520000 | 5 | 1 | channel | ✓ |
| 312 | 679477248 | 20971520000 | 5 | 2 | channel | ✓ |
| 313 | 679477248 | 20971520000 | 6 | 1 | channel | ✓ |
| 314 | 679477248 | 52428800000 | 0 | 7 | channel | ✓ |
| 315 | 679477248 | 52428800000 | 1 | 1 | 32 | ✓ |
| 316 | 679477248 | 52428800000 | 1 | 1 | 64 | ✓ |
| 317 | 679477248 | 52428800000 | 1 | 1 | 128 | ✓ |
| 318 | 679477248 | 52428800000 | 1 | 1 | 256 | ✓ |
| 319 | 679477248 | 52428800000 | 1 | 1 | 512 | ✓ |
| 320 | 679477248 | 52428800000 | 1 | 1 | channel | ✓ |
| 321 | 679477248 | 52428800000 | 1 | 1 | tensor | ✓ |
| 322 | 679477248 | 52428800000 | 1 | 2 | channel | ✓ |
| 323 | 679477248 | 52428800000 | 1 | 3 | channel | ✓ |
| 324 | 679477248 | 52428800000 | 1 | 4 | channel | ✓ |
| 325 | 679477248 | 52428800000 | 1 | 5 | channel | ✓ |
| 326 | 679477248 | 52428800000 | 1 | 6 | channel | ✓ |
| 327 | 679477248 | 52428800000 | 2 | 1 | channel | ✓ |
| 328 | 679477248 | 52428800000 | 2 | 3 | channel | ✓ |
| 329 | 679477248 | 52428800000 | 3 | 1 | channel | ✓ |
| 330 | 679477248 | 52428800000 | 3 | 2 | channel | ✓ |
| 331 | 679477248 | 52428800000 | 4 | 1 | channel | ✓ |
| 332 | 679477248 | 52428800000 | 4 | 3 | channel | ✓ |
| 333 | 679477248 | 52428800000 | 4 | 5 | channel | ✓ |
| 334 | 679477248 | 52428800000 | 5 | 1 | channel | ✓ |
| 335 | 679477248 | 52428800000 | 5 | 2 | channel | ✓ |
| 336 | 679477248 | 52428800000 | 6 | 1 | channel | ✓ |
| 337 | 679477248 | 104857600000 | 0 | 7 | channel | ✓ |
| 338 | 679477248 | 104857600000 | 1 | 1 | 32 | ✓ |
| 339 | 679477248 | 104857600000 | 1 | 1 | 64 | ✓ |
| 340 | 679477248 | 104857600000 | 1 | 1 | 128 | ✓ |
| 341 | 679477248 | 104857600000 | 1 | 1 | 256 | ✓ |
| 342 | 679477248 | 104857600000 | 1 | 1 | 512 | ✓ |
| 343 | 679477248 | 104857600000 | 1 | 1 | channel | ✓ |
| 344 | 679477248 | 104857600000 | 1 | 1 | tensor | ✓ |
| 345 | 679477248 | 104857600000 | 1 | 2 | channel | ✓ |
| 346 | 679477248 | 104857600000 | 1 | 3 | channel | ✓ |
| 347 | 679477248 | 104857600000 | 1 | 4 | channel | ✓ |
| 348 | 679477248 | 104857600000 | 1 | 5 | channel | ✓ |
| 349 | 679477248 | 104857600000 | 1 | 6 | channel | ✓ |
| 350 | 679477248 | 104857600000 | 2 | 1 | channel | ✓ |
| 351 | 679477248 | 104857600000 | 2 | 3 | channel | ✓ |
| 352 | 679477248 | 104857600000 | 3 | 1 | channel | ✓ |
| 353 | 679477248 | 104857600000 | 3 | 2 | channel | ✓ |
| 354 | 679477248 | 104857600000 | 4 | 1 | channel | ✓ |
| 355 | 679477248 | 104857600000 | 4 | 3 | channel | ✓ |
| 356 | 679477248 | 104857600000 | 4 | 5 | channel | ✓ |

| 357 | 679477248 | 104857600000 | 5 | 2 | channel | ✓ |
| 358 | 679477248 | 104857600000 | 6 | 1 | channel | ✓ |
| 359 | 1233125376 | 10485760000 | 1 | 2 | 512 | ✗ |
| 360 | 1233125376 | 10485760000 | 4 | 3 | 512 | ✗ |
| 361 | 1233125376 | 20971520000 | 1 | 2 | 512 | ✗ |
| 362 | 1233125376 | 20971520000 | 4 | 3 | 512 | ✗ |
| 363 | 1233125376 | 52428800000 | 1 | 2 | 512 | ✗ |
| 364 | 1233125376 | 52428800000 | 4 | 3 | 512 | ✗ |
| 365 | 1233125376 | 104857600000 | 1 | 2 | 512 | ✗ |
| 366 | 1233125376 | 104857600000 | 4 | 3 | 512 | ✗ |
| 367 | 7083130880 | 10485760000 | 4 | 3 | 64 | ✗ |
| 368 | 7083130880 | 104857600000 | 4 | 3 | 64 | ✗ |
| 369 | 71236059136 | 10485760000 | 4 | 3 | 64 | ✗ |
| 370 | 71236059136 | 20971520000 | 4 | 3 | 64 | ✗ |

