# OpenReview forum: "Scaling Laws for Floating–Point Quantization Training"
_ICML.cc/2025/Conference — ICML 2025 poster_

### Official Review · Reviewer_EAzc · 2025-03-04

**Overall Recommendation:** 3

**Summary:**

This paper constructs scaling laws for floating point quantized training based on curve fitting to many small to medium scale LLM training experiments.

**Claims And Evidence:**

This paper seems to be mostly based on empirical curve fitting, as most scaling law papers are.

**Essential References Not Discussed:**

The key related work seems to be https://arxiv.org/abs/2411.04330 and the authors do discuss it.

**Experimental Designs Or Analyses:**

See above.

**Methods And Evaluation Criteria:**

This paper seems to make reasonable choices in the context of existing LLM scaling law papers.

**Other Comments Or Suggestions:**

I'm not sure if Comic Sans is against the style guide rules but you clearly should have used Papyrus instead https://www.youtube.com/watch?v=jVhlJNJopOQ.

**Other Strengths And Weaknesses:**

As mentioned above, I think it is hard to evaluate scaling law papers in general. Regarding this specific paper, integer quantization is a special case of EeMm where e = 0. Do your scaling laws reduce to existing integer training scaling laws when e = 0? Likewise, the SNR of EeMm datatypes is relatively constant ignoring over/underflow. Would it be more reasonable to fit a curve to SNR or datatype distortion instead of E and M separately? Figure 5 and the corresponding part of the scaling law suggests that the optimal setup for EeMm is when e == m. How much of this is dependent on activation and weight distributions? Some PTQ and QAT papers have proposed using random orthogonal transformations to transform the activation and weight distributions prior to quantization -- would that change the scaling law?

**Questions For Authors:**

See above

**Relation To Broader Scientific Literature:**

In my opinion, scaling law papers are hard to evaluate in general. On one hard, the results are probably useful, but there does not appear to be a lot of theory behind why models scale as such. The models studied in this work are also small (<1B params), making it hard to judge how much extrapolation is possible with the derived scaling curves. The authors do run validation on a 1.2B parameter model, but 1.2B parameters is still much smaller than production language models that are trained on trillions of tokens (vs. O(100B tokens) in this paper). I don't fault the authors for this since training enough LLMs to fit curves is extremely expensive, but at the same time the authors chose to write a scaling laws paper.

**Theoretical Claims:**

This is a mostly empirical paper.

---

> ### Author Rebuttal · Authors · 2025-03-31
>
> We sincerely thank you for your constructive suggestions and valuable comments! We hope our rebuttal could help address your concerns, and we would be grateful if you could consider increasing the overall recommendation of our work.
>
> ## Q1: Extension to larger models.
> A1: Thanks for the suggestion. We agree with the reviewer that experiments on larger models could further verify the effectiveness of our scaling law. We attempt to answer from the following aspects:
>
> (1) The scaling law exploration of LLMs is essential but extremely expensive in both GPUs and running time. To thoroughly explore the relationships between the loss and different floating-point quantization training factors (e.g., N, D, E, M, B), we have trained 366 models with different settings (model sizes from 41M to 679M) to draw our scaling laws, and successfully validate them on 1.2B models with 8 different settings in Fig. 4. **The adopted model sizes are comparable with those in other scaling law works** (e.g., Scaling law for precision [1], one of the most related work, adopts model sizes up to 1.7B parameter for validation).
>
> (2) We conduct several additional experiments with model sizes larger than 1.2B. Precisely, **we successfully predict the loss of 7B and 70B LLMs (with different settings) based on our scaling law**. The detailed model setting, actual loss and predicted loss are shown as follows:
>
> |N|D|B|E|M|$L_{\text{actual}}$|$L_{\text{predict}}$|$\Delta L$
> |-:|-:|-:|-:|-:|-:|-:|-:|
> |1.2B|100B|512|4|3|2.50|2.54|-0.04
> |7B|10B|64|4|3|2.65|2.70|-0.05
> |7B|100B|64|4|3|2.38|2.38|0.00
> |70B|10B|64|4|3|2.60|2.56|0.04
> |70B|20B|64|4|3|2.44|2.42|0.02
>
> (The results of 1.2B models have already been included in Fig. 4 of our paper)
>
> The actual losses of 7B/70B models are very close to the theoretical value calculated by our scaling law. Therefore, we could confidently claim that our scaling law could extend beyond larger model sizes. These additional results will also be given in the revision.
>
> ## Q2: Do the scaling laws reduce to existing integer training scaling laws when e = 0?
>
> A2: We thank the reviewer for this insight. Mathematically, our scaling laws approximately reduce to integer quantization formulations when e=0. However, there exist fundamental differences in hardware implementation between integer (fixed-point) and floating-point arithmetic: integer operations rely on dedicated fixed-point units and two’s complement representation, while floating-point architectures require separate processing of sign, exponent, and mantissa bits through distinct computational mechanisms. Such hardware-level disparities suggest that theoretical simplifications may not directly translate to empirical training scenarios. Our current work focuses on floating-point quantization scaling laws ($e \neq 0$), while the formal alignment of the special e=0 integer case with existing theories will be explored as a future direction. This requires joint optimization analysis incorporating hardware instruction sets and numerical representation properties for rigorous empirical validation. We will add this discussion in our revision.
>
> ## Q3: Would it be more reasonable to fit a curve to SNR or datatype distortion instead of E and M separately? How much of this is dependent on activation and weight distributions? Would some PTQ and QAT papers with random orthogonal transformations change the scaling law?
>
> A3: (1) This is a valuable suggestion. As shown in Eq. 37 of  [1], using the Signal-to-Quantization-Noise Ratio (SQNR):
> $$\text{SQNR} = 10 \log_{10}\left(
>     \frac{
>         \mathbb{E}[\mathbf{W}^2]\mathbb{E}[\mathbf{X}^2]
>     }{
>         \mathbb{E}[(\mathbf{W}\mathbf{X} - \mathcal{Q}(\mathbf{W})\mathcal{Q}(\mathbf{X}))^2]
>     }
> \right)$$
> for GEMM operations could unify the effects of exponent (E) and mantissa (M) precision while accounting for input distribution modifications like random orthogonal transformations. However, SQNR calculation inherently depends on tensor distributions, which may limit its practical applicability. Therefore, we directly select the raw exponent and mantissa bits rather than the SNR as essential factors in our scaling law for more precise prediction ability.
>
> (2) Notably, our theory explicitly models dataset size (D) as a variable, and since weight/activation distributions evolve during training, we believe that our formulation exhibits partial robustness to such distributional shifts. The interaction between PTQ/QAT methods and scaling laws remains an open question requiring systematic analysis of how orthogonal transformations alter quantization noise dynamics.  We will add our discussions in revision.
>
> ## Q4: Format and references.
>
> A4: Thanks for your suggestion. We will fix them in revision.
>
> ## References:
>
> [1] Kuzmin, Andrey et al. FP8 Quantization: The Power of the Exponent.

---

> > ### Comment · Reviewer_EAzc · 2025-04-04
> >
> > Thank you for your response and additional experiments. I will keep my score for now.

---

### Official Review · Reviewer_sLff · 2025-03-08

**Overall Recommendation:** 3

**Summary:**

This paper proposes a scaling law for LLM performance prediction according to model size, dataset size, exponent bit, and mantissa bit while training LLM under FP quantization. Based on previous research, the paper tries to predict LLM performance more precisely. To achieve this objective, the paper proposes the following.

1. Exponent bit is of greater importance than mantissa bit.
2. While training LLM under low precision conditions, the excessive size of a dataset which is larger than its critical size can degrade the performance of the model.
3. The optimal balance between cost and performance is among 4~8 bits.

**Claims And Evidence:**

- Exponent bit is more important than mantissa bit.
    - The paper supports this claim with various experiments and provides optimized exponent bit and mantissa bit settings per total number of bits.
    - There are already several works that contend the importance of exponent bits, so it is not a novel idea.
- There is a critical size of a dataset according to the size of a model.
    - Figure 6 is depicted with not actual loss but predict loss. However, there isn’t any actual performance validation with benchmark datasets. It is not appropriate to draw conclusions based solely on predict loss value, without considering validation performance with real data.

**Essential References Not Discussed:**

`FP8 Quantization: The Power of the Exponent` published in Neurips 2022 already addressed the importance of exponent bit while FP quantization.

**Experimental Designs Or Analyses:**

The authors provide graphs about the correlation between predict loss and actual loss to verify how the proposed scaling law regresses the performance of language models well. The proposed scaling law shows better results compared to the previous works. However, they don’t validate the scaling law with the target models and practical validation datasets.
It would be better to compare the correlation between the outputs of the scaling law and validation datasets.
Likewise, verifying the claims with predict loss only is not persuasive, thus additional supports are needed.

**Methods And Evaluation Criteria:**

- The proposed scaling law
    - The authors show that the prediction of the proposed scaling law is more accurate compared to previous works with several experiments.
    - However, the paper doesn’t clearly provide how the constants in the scaling law equations are determined.
    - Also, there aren’t any experiments with benchmark datasets.
- Propose optimal float layout per bit-width
    - The authors find optimal values with the proposed scaling law.
    - The grounds for asserting that the proposed layout is optimal are weak because the authors don’t provide experiments with any benchmark datasets.

**Other Comments Or Suggestions:**

The paper refers to figures like `Figure N`, but Figure 6 is referred to as `Fig 6`. It would be better to unify the reference style.

**Other Strengths And Weaknesses:**

As the authors already mentioned in the paper, performance prediction of LLMs according to quantization bit settings can help design hardware architecture well.
Also, from a practical perspective, if we can predict the size of the model and the amount of data for achieving the desired performance, it will help reduce costs.
Therefore, this research topic has good scalability.
However, the paper’s claims demonstrate limited novelty. The importance of the exponent bit for LLM is already known. Without having to the paper mentioned above, BF16 which has a larger number of exponent bits compared to FP16 is widely used for LLM.
Also, the support of the claims is insufficient. The authors propose the scaling law, but they don’t provide how the scaling law is induced and how the constants in the scaling law are determined. Moreover, the authors verify claims based on predict loss, after showing that the predict loss calculated with the proposed scaling law can approximate actual loss well. However, it is not persuasive because the paper doesn’t confirm the claims with actual loss or validation performance with real datasets

**Questions For Authors:**

- Figure 5 shows optimized bit settings between exponent and mantissa according to total bit-width. However, the authors don’t use those settings in Figure 6. Does a similar pattern appear based on the settings obtained from Figure 5?
- In experiment settings in the Appendix, the size of target models is significantly smaller than those of current interest. Does the proposed scaling law hold strongly for larger models and various models with marginal differences?

**Relation To Broader Scientific Literature:**

It has already been addressed in previous works that the exponent bit is important during FP quantization. Thus their claim is not a novel finding.

**Theoretical Claims:**

The authors explain the proposed scaling law with mathematical formulas. However, the authors just assume that those formulas are right, and there isn’t rationale for how to induce them.

---

> ### Author Rebuttal · Authors · 2025-03-31
>
> We sincerely thank you for your constructive suggestions and valuable comments! We hope our rebuttal could help address your concerns. If so, we would be grateful if you could consider increasing the overall recommendation of our work.
>
> ## Q1: Novelty. Several works have already emphasized the importance of exponent bits.
>
> A1: (1) The central contribution of this work is to build a **scaling law for floating-point quantization training, which quantitatively models the relationships between the loss L and several essential factors**, e.g., data size D, model size N, exponent E, mantissa M, and block size of scaling factors B, by providing a joint scaling law formation in Eq. 1. We are the **first** to give this scaling law. We highlight that the discussions on the importance of exponent bit and mantissa bit are just one part of our multiple contributions, and **the novelty of our work still holds**.
>
> (2) Although there are some efforts that have explored the importance of exponent bit while FP quantization (e.g., FP8 Quantization: The Power of the Exponent), our efforts and discussions on exponent and mantissa within the scaling law are still novel and valuable: (a) Different from existing works, our work firstly provides a scaling law that could **quantitatively predict the model loss** via D, N, E, M, B, rather than simply focusing on the importance of E and M. Our work enables accurate loss prediction of LMs in practice, verifying the importance from a quantitative aspect. (b) Besides the quantitative marginal effects of exponent and mantissa, we further explore the quantitative **joint effects of D, N, E, M, B** on model loss.
>
> (3) Besides the novel findings on E and M, we also discover several **valuable observations and insights with extensive experiments** that could facilitate future low-precision LLM training in the last paragraph of Sec. 1.
>
> We will add this discussion and related works in our revision.
>
> ## Q2: Evaluation setting. The authors don’t provide experiments with benchmark datasets.
>
> A2: (1) Thanks for the suggestion. This work concentrates on providing a scaling law for FP quantization training. Following previous works on scaling laws [1,2] (including [1] for INT quantization), we conduct extensive experiments with different precision settings (366 models) to build the basic scaling law formation of various D/N/E/M/B, and validate its effectiveness on larger models. In Fig. 4, we have clearly shown the good fitting results of our scaling law. The validation points (1.2B models, noted as yellow stars) in the bottom left corner fit well for our scaling law, implying that the scaling law could extend to larger models (the fitting results of 7B/70B models are also satisfactory given in rebuttal). We DO NOT draw conclusions based solely on predicted loss value, but **rely on the fitting results of predicted losses and actual losses on real data**.
>
> (2) The goal of scaling law is to predict the loss based on essential factors. Therefore, it is natural and best to evaluate the correctness based on the differences between actual losses and predicted losses given by our scaling law. [1] is the most related work that also adopts the same evaluation metric fitting results for validation and does not evaluate on downstream benchmarks. We also clarify that the loss is a practical and accurate indicator that provides an overall evaluation of LMs' capabilities widely used in the real world, which is strongly correlated to the overall performance on downstream tasks.
> We will add this discussion in our revision.
>
> ## Q3: How are scaling law equation constants determined?
>
> A3: We have introduced in Sec. 3 and 4.1 that we first conduct experiments to reveal the marginal effects, and then design the joint scaling law. The formations are designed based on empirical insights and the distributions of losses in 366 experiments. Next, we fit and validate the formation via LM settings (D/N/E/M/B) and the corresponding losses to obtain the constants in our scaling laws. These constants are learned from practical losses, not theoretical derivation, similar to other works [1,2].
>
> ## Q4: Whether optimized bit settings have a similar pattern in Fig. 6?
>
> A4: As stated in Sec. 4.3, Fig. 6 displays the implication that there is an optimal data size under a certain FP quantization setting **theoretically based on our scaling law**. The validation of our scaling law is in Sec. 4.1 and Fig. 4 (as in A2). Fig. 6 aims to show the intuitive phenomenon of theoretical “optimal data size” in different settings. Hence, using the optimized bit settings in Fig. 5 also has a similar pattern due to the formation. A comprehensive derivation of $D_{crit}$ is in Appendix D.
>
> ## Q5: Extension to larger models.
>
> A5: Due to space limits, please refer to Reviewer UF7C's A1 for larger LMs' results (7B/70B).
>
> ## References
>
> [1] Kumar T et al. Scaling laws for precision.
>
> [2] Hoffmann J et al. Training compute-optimal large language models.

---

### Official Review · Reviewer_UF7C · 2025-03-09

**Overall Recommendation:** 3

**Summary:**

The paper proposes a new scaling law tailored specifically for floating-point quantization during training of large language models (LLMs). Authors extensively studied how quantization parameters—exponent bits, mantissa bits, and scaling block sizes—impact LLM performance. Through extensive empirical experiments (366 runs), they developed a unified scaling law for predicting LLM losses under floating-point quantization. The authors also identified optimal bit layouts, critical data sizes to prevent performance degradation under low precision, and determined that 4–8 bits offer the best trade-off between computational cost and model performance.

## update after rebuttal
I maintain my original score. I am generally satisfied with the authors’ response.

**Claims And Evidence:**

- Authors claim that exponent bits matter a bit more than mantissa bits for performance. This was supported by many experiments showing lower loss with optimal exponent allocation.
- They further claim that there's a critical data size, beyond which adding extra data actually hurts performance in low precision. Authors mathematically derived this critical point and validated empirically across different configurations.
- Finally, authors claim that optimal quantization precision depends on computational resources available, best balance typically between 4 to 8 bits. Experiments over many model sizes, precision levels, and compute budgets repeatedly show this optimal precision window.

**Essential References Not Discussed:**

Authors ignored real hardware studies like [1], [2] [3], who provide significant insights into floating-point performance on actual hardware. Including these studies would make findings stronger and more relevant.

[1] Kuzmin, Andrey et al. “FP8 Quantization: The Power of the Exponent.” ArXiv abs/2208.09225 (2022).
[2] Baalen, Mart van et al. “FP8 versus INT8 for efficient deep learning inference.” ArXiv abs/2303.17951 (2023).
[3] Aggarwal, Shivam et al. “Shedding the Bits: Pushing the Boundaries of Quantization with Minifloats on FPGAs.” 2024 34th International Conference on Field-Programmable Logic and Applications (FPL) (2023): 297-303.

**Experimental Designs Or Analyses:**

Experiments were thorough and well-organized, systematically exploring multiple dimensions: exponent/mantissa combinations, data scales, block sizes, and model sizes. Yet, the largest model studied was only 1.2B parameters, somewhat limiting how confidently we can extrapolate findings to the extremely large models popular nowadays (e.g., tens or hundreds of billions).

**Methods And Evaluation Criteria:**

The authors used Transformer-based LLMs trained on subsets of the Dolma dataset, with sizes from 41M to 1.2B parameters. Quantization was simulated via QPyTorch, carefully varying exponent, mantissa, block size, data, and model sizes. Empirical outcomes were systematically compared with existing scaling laws (Chinchilla, OpenAI and Kumar et al. 2024), highlighting improvements over prior work.

**Other Comments Or Suggestions:**

Line 122  "Current Scaling Laws cannot Well Fit in Floating-point Quantization" can be rephrased -> Current Scaling Laws cannot Fit Well in Floating-point Quantization"

**Other Strengths And Weaknesses:**

**Strengths**
- Clearly fills a gap by focusing specifically on floating-point quantization.
- Robust empirical validation with extensive experiments.
- Offers actionable guidance on optimal floating-point bit allocation.
- Presents clear visualizations (especially Figure 5 on optimal bit layouts) that make insights easy to understand.

**Weaknesses**
- Experiments capped at relatively modest model scales (max ~1B).
- Hardware validation is missing, which could impact the practical usability of proposed methods.

**Questions For Authors:**

- Do you think the proposed scaling law can extend beyond model sizes > 1.2 B?
- Can the observed relationships between exponent and mantissa bit precision in your paper be conceptually related or compared with the notion of an 'effective parameter count' introduced by Kumar et al. (2024)? Would integrating such a concept clarify or enhance your theoretical interpretation?

**Relation To Broader Scientific Literature:**

The authors cited foundational scaling law papers (e.g., Kaplan et al. 2020; Hoffmann et al. 2022; Kumar et al. 2024) adequately.

**Theoretical Claims:**

The scaling law is empirically derived by running a comprehensive series of experiments (366 runs) varying parameters like model size, data size, exponent bits, mantissa bits, and block sizes. Other existing works like Kaplan et al. (2020), Hoffmann et al. (Chinchilla law), and Kumar et al. (2024), are also predominantly empirically derived.

---

> ### Author Rebuttal · Authors · 2025-03-31
>
> We sincerely thank you for your constructive suggestions and valuable comments! We hope our rebuttal could help address your concerns, and we would be grateful if you could consider increasing the overall recommendation of our work.
>
> ## Q1: Experiments capped at relatively modest model scales (max ~1B). Do you think the proposed scaling law can extend beyond model sizes > 1.2 B?
>
> A1: Thanks for the suggestion. We agree with the reviewer that experiments on larger models could further verify the effectiveness of our scaling law. We attempt to answer from the following aspects:
>
> (1) The scaling law exploration of LLMs is essential but extremely expensive in both GPUs and running time. To thoroughly explore the relationships between the loss and different floating-point quantization training factors (e.g., N, D, E, M, B), we have trained 366 models with different settings (model sizes from 41M to 679M) to draw our scaling laws, and successfully validate them on 1.2B models with 8 different settings in Fig. 4. **The adopted model sizes are comparable with those in other scaling law works** (e.g., Scaling law for precision [1], one of the most related work, adopts model sizes up to 1.7B parameter for validation).
>
> (2) We conduct several additional experiments with model sizes larger than 1.2B. Precisely, **we successfully predict the loss of 7B and 70B LLMs (with different settings) based on our scaling law**. The detailed model setting, actual loss and predicted loss are shown as follows:
>
> |N|D|B|E|M|$L_{\text{actual}}$|$L_{\text{predict}}$|$\Delta L$
> |-:|-:|-:|-:|-:|-:|-:|-:|
> |1.2B|100B|512|4|3|2.50|2.54|-0.04
> |7B|10B|64|4|3|2.65|2.70|-0.05
> |7B|100B|64|4|3|2.38|2.38|0.00
> |70B|10B|64|4|3|2.60|2.56|0.04
> |70B|20B|64|4|3|2.44|2.42|0.02
>
> (The results of 1.2B models have already been included in Fig. 4 of our paper)
>
> The actual losses of 7B/70B models are very close to the theoretical value calculated by our scaling law. Therefore, we could confidently claim that our scaling law could extend beyond larger model sizes. These additional results will also be given in the revision.
>
> ## Q2: Hardware validation is missing, which could impact the practical usability of proposed methods.
>
> A2: Thanks for your suggestion. We agree that these related works can provide valuable insights of floating-point quantization in practice. We will add these noted related works in our revision with detailed discussions.
>
> ## Q3: Can the observed relationships between exponent and mantissa bit precision in your paper be conceptually related or compared with the notion of an 'effective parameter count' introduced by [1]? Would integrating such a concept clarify or enhance your theoretical interpretation?
>
> A3: Thanks for the suggestion. As discussed in Appendix M of [1], the effective parameter count is formulated as a counterpart to the parameter count N in the Chinchilla scaling law. In Appendix C, Eq. 37 of our paper, we derive an analogous concept where our equivalent N demonstrates not only a positive correlation with precision metrics: $P^{\delta + \nu}$ or $(E + 0.5)^\delta(M + 0.5)^\nu$, aligning with their framework, but also a negative correlation with dataset size D. This reveals that increasing training data volume effectively reduces the equivalent parameter count, implying that larger datasets amplify the impact of numerical precision on model expressivity.
>
> ## References:
>
> [1] Kumar T et al. Scaling laws for precision.

---

### Decision · Program_Chairs · 2025-05-01

**Decision:**

Accept (poster)

**Comment:**

The paper proposes a more fine-grained scaling law for quantized training, explicitly modeling the exponent bit and the mantissa bit instead of merging them as done in Kumar et al. This is shown to give a better fit due to the constituents' different roles and the paper provides an optimal bit ratio. The reviewers find the paper to be a well-executed empirical study and the findings potentially useful, with some reservations on novelty (in light of Kumar et al. and prior non-scaling law works that already show the relative importance of the exponent bit) and relatively small-scale experiments.